# Integrated analysis of environmental and genetic influences on cord blood DNA methylation in new-borns

Darina Czamara ⓘ et al.[#]

Epigenetic processes, including DNA methylation (DNAm), are among the mechanisms allowing integration of genetic and environmental factors to shape cellular function. While many studies have investigated either environmental or genetic contributions to DNAm, few have assessed their integrated effects. Here we examine the relative contributions of prenatal environmental factors and genotype on DNA methylation in neonatal blood at variably methylated regions (VMRs) in 4 independent cohorts (overall $n = 2365$). We use Akaike's information criterion to test which factors best explain variability of methylation in the cohort-specific VMRs: several prenatal environmental factors (E), genotypes in cis (G), or their additive (G + E) or interaction (GxE) effects. Genetic and environmental factors in combination best explain DNAm at the majority of VMRs. The CpGs best explained by either G, G + E or GxE are functionally distinct. The enrichment of genetic variants from GxE models in GWAS for complex disorders supports their importance for disease risk.

F oetal or prenatal programming describes the process by which environmental events during pregnancy influence the development of the embryo with on-going implications for future health and disease. Several studies have shown that the in utero environment is associated with disease risk, including coronary heart disease[1,2], type 2 diabetes[3], childhood obesity[4,5] as well as psychiatric problems[6] and disorders[7–9].

Environmental effects on the epigenome, for example, via DNA methylation, could lead to sustained changes in gene transcription and thus provide a molecular mechanism for the enduring influences of the early environment on later health[10]. Smoking during pregnancy influences widespread and highly reproducible differences in DNA methylation at birth[11]. Less dramatic effects have been reported for maternal body mass index (BMI)[12], pre-eclampsia and gestational diabetes[13,14]. Possible epigenetic changes as a consequence of prenatal stress are less well established[15]. Some of these early differences in DNA methylation persist, although attenuated, through childhood[11,16] and might be related to later symptoms and indicators of disease risk, including BMI during childhood[17,18] or substance use in adolescence[19]. These data emphasise the potential importance of the prenatal environment for the establishment of inter-individual variation in the methylome as a predictor or even mediator of disease risk trajectories.

In addition to the environment, the genome plays an important role in the regulation of DNA methylation. To this end, the impact of genetic variation, especially of single nucleotide polymorphisms (SNPs) on DNA methylation in different tissues, has resulted in the discovery of a large number of methylation quantitative trait loci (meQTLs, i.e., SNPs significantly associated with DNA methylation status[20]). These variants are primarily in cis, i.e., at most 1 million base pairs away from the DNA methylation site[20–22] and often co-occur with expression QTLs or other regulatory QTLs[23–25]. The association of meQTLs with DNA methylation is relatively stable throughout the life course[21]. In addition, SNPs within meQTLs are strongly enriched for genetic variants associated with common disease in large genome-wide association studies (GWAS) such as BMI, inflammatory bowel disease, type 2 diabetes or major depressive disorder[21,23,24,26].

Environmental and genetic factors may act in an additive or multiplicative manner to shape the epigenome to modulate phenotype presentation and disease risk[27]. However, very few studies have so far investigated the joint effects of environment and genotype on DNA methylation, especially in a genome-wide context. Klengel et al.[28], for instance, reported an interaction of the FK506 binding protein 5 gene (FKBP5) SNP genotype and childhood trauma on FKBP5 methylation levels in peripheral blood cells, with trauma associated changes only observed in carriers of the rare allele. The most comprehensive study of integrated genetic and environmental contributions to DNA methylation so far was performed by Teh et al.[29]. This study examined variably methylated regions (VMRs), defined as regions of consecutive CpG-sites showing the highest variability across all methylation sites assessed on the Illumina Infinium Human-Methylation450 BeadChip array. In a study of 237 neonate methylomes derived from umbilical cord tissue, the authors explored the proportions of the influence of genotype vs. prenatal environmental factors such as maternal BMI, maternal glucose tolerance and maternal smoking on DNA methylation at VMRs. They found that 75% of the VMRs were best explained by the interaction between genotype and environmental factors (GxE) whereas 25% were best explained by SNP genotype and none by environmental factors alone. Collectively, these studies highlight the importance of investigating the combination of environmental and genetic contributions to DNA methylation and not only their individual contribution.

The main objective of the present study is to extend our knowledge of combined effects of prenatal environment and genetic factors on DNA methylation at VMRs. Specifically, this is addressed by: (1) assessing the stability of the best explanatory factors across different cohorts and whether this extends to all environmental factors, (2) dissecting differences between additive and interactive effects of gene and environment not explored in Teh et al., (3) testing whether VMRs influenced by genetic and/or environmental factors might have a different predicted impact on gene regulation and (4) evaluating the relevance of genetic variants that interact with the environment to shape the methylome for their contribution to genetic disease risk.

Our results show that across cohorts genetic variants in combination with prenatal environment are the best predictors of variance in DNA methylation. We observe functional differences of both the genetic variants and the methylation sites best explained by genetic or additive and interactive effects of genes and environment. Finally, the enrichment of genetic variants within additive as well as interactive models in GWAS for complex disorders supports the importance of these environmentally modified methylation quantitative trait loci for disease risk.

## Results

**Cohorts and analysis plan**. We investigated the influence of the prenatal environment and genotype on VMRs in the DNA of 2365 newborns within 4 different cohorts: Prediction and Prevention of Pre-eclampsia and Intrauterine Growth Restrictions (PREDO, cordblood)[30], the UCI cohort (refs. [31–33], heel prick), the Drakenstein Child Health Study (DCHS, cordblood)[34,35] and the Norwegian Mother and Child Cohort Study (MoBa, cord-blood[36]). A description of the workflow of this manuscript is given in Fig. 1 and the details for each of the cohorts are given in Table 1.

We analysed 963 cord blood samples from the PREDO cohort with available genome-wide DNA methylation and genotype data. Of these samples, 817 had data on the Illumina 450k array (PREDO I) and 146 on the Illumina EPIC array (PREDO II). The main analyses are reported for PREDO I, and replication and extension of the results is shown for PREDO II as well as for three independent cohorts including 121 heel prick samples (UCI cohort, EPIC array) as well as 258 (DCHS, 450 K and EPIC array) and 1023 cord blood samples (MoBa, 450 K array). We tested 10 different prenatal environmental factors covering a broad spectrum of prenatal phenotypes (see Table 1) (referred to as E), as well as cis SNP genotype (referred to as G), i.e., SNPs located in at most 1MB distance to the specific CpG, additive effects of cis SNP genotype and prenatal environment (G + E) and cis SNP×environment interactions (GxE) for association with DNA methylation levels (see Fig. 1). We then assessed for each VMR independently which model described the variance of DNAm best using Akaike's information criterion (AIC)[37]. In all models, we corrected for child's gender, ethnicity (using MDS-components), gestational age as well as estimated cell proportions to account for cellular heterogeneity.

**Variably methylated regions**. We first identified candidate VMRs, defined as regions of CpG-sites showing the highest variability across all methylation sites. In PREDO I, we identified 10,452 variable CpGs that clustered into 3982 VMRs (see Supplementary Data 1). Most VMRs ($n = 2683$) include 2 CpGs. As detailed in Supplementary Note 1, the distribution of methylation levels of CpGs within these VMRs is unimodal, (see Supplementary Fig. 1A), VMRs are enriched in specific functional regions of the genome, correlate with differences in gene

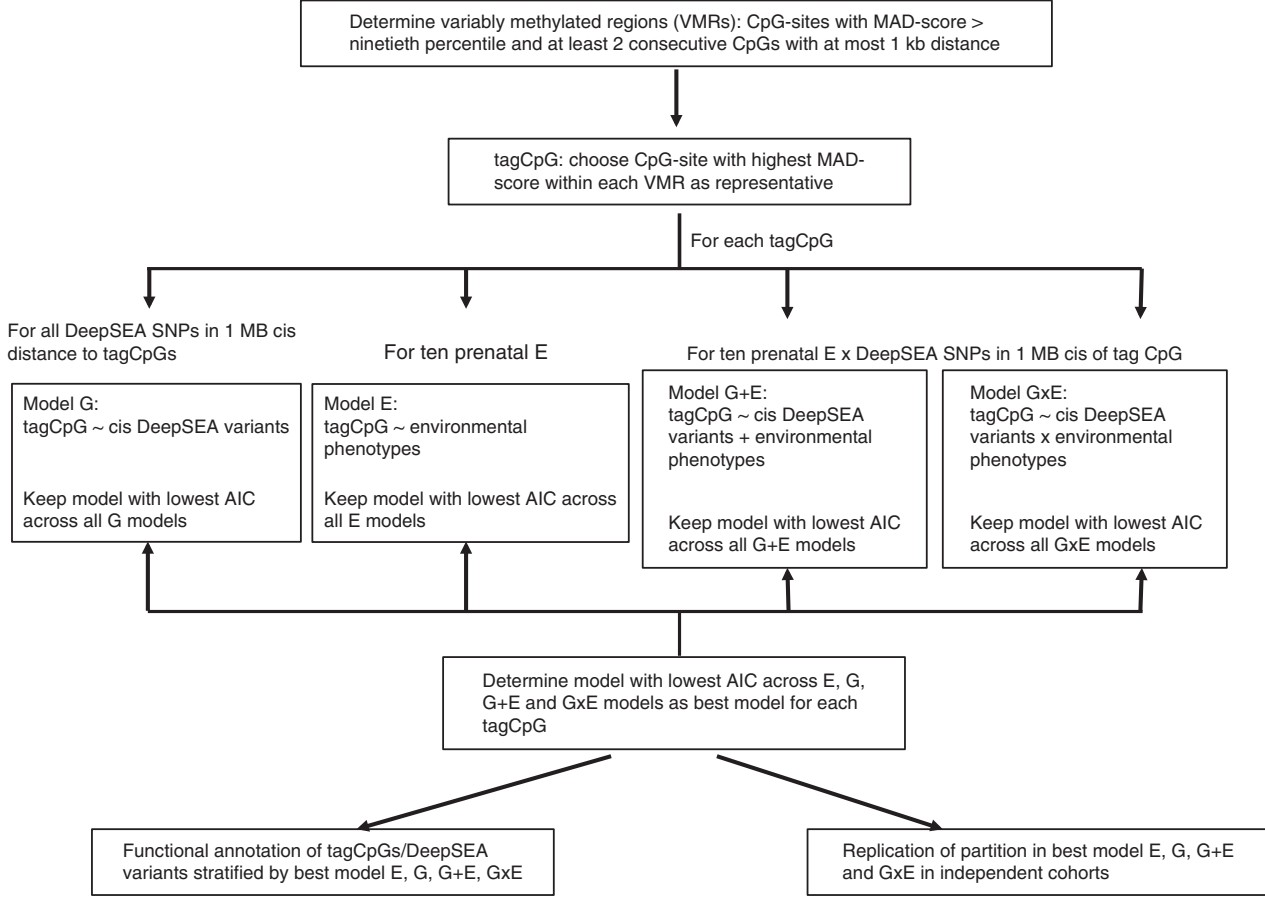

**Fig. 1** Flow diagram of VMR analysis

expression, and overlap with sites associated with specific prenatal environmental factors.

To examine the factors that best explain the variance in methylation in these functionally relevant sites, we chose the CpG-site with the highest MAD score as representative of the VMR. These CpGs are named tagCpGs. The correlation between methylation levels of tagCpG and average methylation of the respective VMR was high (mean $r = 0.85$, sd $r = 0.08$), suggesting that the tag CpGs are valid representatives of their VMRs. Furthermore, tagCpGs are mainly uncorrelated with each other (mean $r = 0.03$, sd $= 0.12$).

**Which models explain methylation of tagCpGs best?** We next compared the fit of four models for each of the 3,982 tagCpGs (see Fig. 1): best SNP (G model), best environment (E model), SNP+ environment (G + E model) and SNP× environment (GxE model). Association results for each model are listed in Supplementary Data 2–5. For each tagCpG, the model with the lowest AIC was chosen as the best model (see Methods section). In total, 40.6% of tagCpGs were best explained by GxE ($n = 1616$), followed by G (30%, $n = 1, 194$) and G + E (29%, $n = 1171$) (Fig. 2a). E explained most variance in one tagCpG. All tag CpGs and the respective SNPs and environments from the best model are listed in Supplementary Data 6–8 and Supplementary Table 1.

With regard to environmental factors, 27.0% of tagCpGs best explained by the G + E model were associated with environmental factors related with stress or, in particular, glucocorticoids (i.e., maternal betamethasone treatment), 40.8% with general maternal factors (mostly maternal age) and 32.20 % with factors related to metabolism (pre-pregnancy BMI, hypertension,

gestational diabetes). For best model GxE tagCpGs, the proportions of environmental factors were similar with 22.2, 44.1 and 33.7%, respectively (see Fig. 2b).

We next looked into the delta AIC, i.e., the difference between the AIC of the best model to the AIC of the next best model (see Supplementary Note 2). GxE models appear to be winning by a significantly larger AIC margin over the next best model, when compared to the other types of winning models (see Fig. 2c).

**DeepSEA prediction of SNP function.** We were next interested in understanding the functionality of both the VMRs as well as the associated SNPs in the G, GxE and G + E models. For this we restricted the analyses only to potentially functional relevant SNPs using DeepSEA[38] and not all linkage disequilibrium (LD)-pruned SNPs as described above. DeepSEA, a deep neural network pretrained with DNase-seq and ChIP-seq data from the ENCODE[39] project, predicts the presence of histone marks, DNase hypersensitive regions (DHS) or TF binding for a given 1 kb sequence. The likelihood that a specific genetic variant influences regulatory chromatin features is estimated by comparing predicted probabilities of two sequences where the bases at the central position are the reference and alternative alleles of a given variant. We reran the four models now restricting the cis-SNPs to those 36,241 predicted DeepSEA variants that were available in our imputed, quality-controlled genotype dataset.

Top results for models including G, GxE and G + E are depicted in Supplementary Data 9–12.

Results were comparable to what we observed before: 1195 (30.09%) of tagCpGs presented with best model G, 1193 CpGs (30.04%) with best model G + E, 1510 CpGs (38.02%) with best

**Table 1 Overview of investigated cohorts**

| Cohort | PREDO I | PREDO II | DCHS I | DCHS II | UCI | MoBa |
|---|---|---|---|---|---|---|
| Sample size | 817 | 146 | 107 | 151 | 121 | 1023 |
| Methylation array | Illumina 450 K | Illumina EPIC | Illumina 450 K | Illumina EPIC | Illumina EPIC | Illumina 450 K |
| Methylation data processing | Funnorm and Combat | Funnorm and Combat | SWAN and Combat | BMIQ and Combat | Funnorm and Combat | BMIQ and Combat |
| SNP genotyping | Illumina Human Omni Express Exome | Illumina Human Omni Express Exome | Illumina PsychArray | Illumina GSA | Illumina Human Omni Express | Illumina HumanExome Core |
| Infant gender male | 433 (53.0%) | 75 (51.4%) | 63 (58.8%) | 83 (55.0%) | 65 (53.7%) | 478 (46.7%) |
| Maternal age mean (sd) | 33.28 (5.79) | 32.25 (4.92) | 26.27 (5.87) | 27.42 (5.93) | 28.47 (4.91) | 29.92 (4.32) |
| Partity mean (sd) | 1.05 (1.02) | 0.87 (1.03) | 0.98 (1.12) | 1.09 (1.07) | 1.11 (1.15) | 0.83 (0.88) |
| Caesarian section | 169 (20.7%) | 36 (24.7%) | 19 (17.6%) | 35 (23.2%) | 37 (30.6%) | 228 (22.3%) |
| Pre-pregnancy BMI mean (sd) | 27.42 (6.40) | 25.37 (5.79) | Not available | Not available | 27.90 (6.44) | 24.05 (4.19) |
| Maternal smoking yes | Exclusion criterion | Exclusion criterion | 7.40 (10.52)[a] | 4.94 (9.43)[a] | 10 (8.2%) | 148 (14.4%) |
| Gestational diabetes yes | 183 (22.4%) | 19 (13.0%) | No cases available | No cases available | 9 (7.4%) | 15 (1.5%) |
| Hypertension yes | 275 (33.7%) | 31 (21.2%) | 2 (0.19%) | 2 (1.3%) | 7 (5.8%) | 50 (4.9%) |
| Betamethasone treatment yes | 35 (4.3%) | 2 (1.5%) | Not available | Not available | No cases available | Not available |
| Anxiety score mean (sd) | 33.93 (7.90)[b] | 34.43 (8.38)[b] | 5.70 (4.15)[c] | 5.32 (3.91)[c] | 1.67 (0.41)[d] | 4.79 (1.36)[e] |
| Depression score mean (sd) | 11.34 (6.47)[f] | 11.53 (6.98)[f] | 17.64 (12.10)[g] | 12.52 (11.55)[g] | 0.68 (0.41)[h] | 5.24 (1.57)[e] |

[a] Based on ASSIST Tobacco Score
[b] STAI sum scores
[c] SRQ-20
[d] STAI average scores
[e] Based on Hopkins Symptom Checklist
[f] CESD sum scores
[g] BDI-II
[h] CESD average score

model GxE and 74 CpGs (1.86%) with best model E (Fig. 3a) and also showed similar differences in delta-AIC and proportions of E categories (see Supplementary Note 3). Only 10 tagCpGs did not present with any DeepSEA variant within 1MB distance in cis and were therefore not further considered. All respective CpG-environment-DeepSea SNP combinations are depicted in Supplementary Data 13–16.

The distribution of best models was not influenced by the degree of variability of DNA methylation, but was comparable across the whole range of DNA methylation variation (see Supplementary Note 4 and Supplementary Fig. 2). A slight enrichment for G + E models was observed in longer VMRs with at least 3 CpGs ($p = 9.00 \times 10^{-06}$, OR = 1.39, Fisher-test, see Supplementary Fig. 3).

In conclusion, also when we focus on potentially functionally relevant SNPs, it is the combination of genotype and environment which best explains VMRs.

We observed that, as expected, different types of exposures or maternal factors have different relative impact on DNA methylation (see Supplementary Note 5). However, even for those exposures with the highest fraction of VMRs best explained by E alone, combined models of G + E and GxE remain the best models in even higher fractions of VMRs (see Supplementary Fig. 4B).

**Functional annotation of different best models.** Focusing on combinations between tagCpGs, environmental factors and DeepSEA variants, we found functional differences for both the SNPs as well as the tagCpGs (see Supplementary Note 6) within the different models. Overall, 895 DeepSEA variants were uniquely involved in best G models, 905 were uniquely in best

G + E models and 1162 uniquely in best GxE models. As a DeepSEA variant can be in multiple 1 MB-cis windows around the tagCpGs, several DeepSEA variants were involved in multiple best models: 138 DeepSEA variants overlapped between G and GxE, 118 between G and G + E and 147 between GxE and G + E VMRs. We observed no significant differences with regard to gene-centric location for DeepSEA variants involved only in G models, only in G + E models or in multiple models. However, DeepSEA variants involved only in GxE models were significantly depleted for promoter locations ($p = 3.92 \times 10^{-02}$, OR = 0.79, Fisher-test, see Supplementary Fig. 5A).

Although no significant differences were present, DeepSEA SNPs involved in the G and G + E model were located in closer proximity to the specific CpG (model G: mean absolute distance = 256.8 kb, sd = 291.2 kb, model G + E: mean absolute distance = 244.8 kb, sd = 284.0 kb, Supplementary Fig. 5B) whereas DeepSEA SNPs involved in GxE models (mean absolute distance = 352.6 kb, sd = 305.3 kb) showed broader peaks around the CpGs.

With regards to histone marks, DeepSEA variants in general were enriched across multiple histone marks indicative of active transcriptional regulation (Fig. 4c). DeepSEA variants involved in best model G + E showed further enrichment for strong transcription ($p = 7.19 \times 10^{-03}$, OR = 1.34, Fisher-test) as well as depletion for quiescent loci ($p = 7.17 \times 10^{-03}$, OR = 0.78, Fisher-test). In contrast, GxE DeepSEA variants were significantly enriched in these regions ($p = 2.62 \times 10^{-02}$, OR = 1.22, Fisher-test, Fig. 4d).

Taken together, these analyses indicate that both the genetic variants and the VMRs in the different best models (G, GxE and G + E) preferentially annotate to functionally distinct genomics regions.

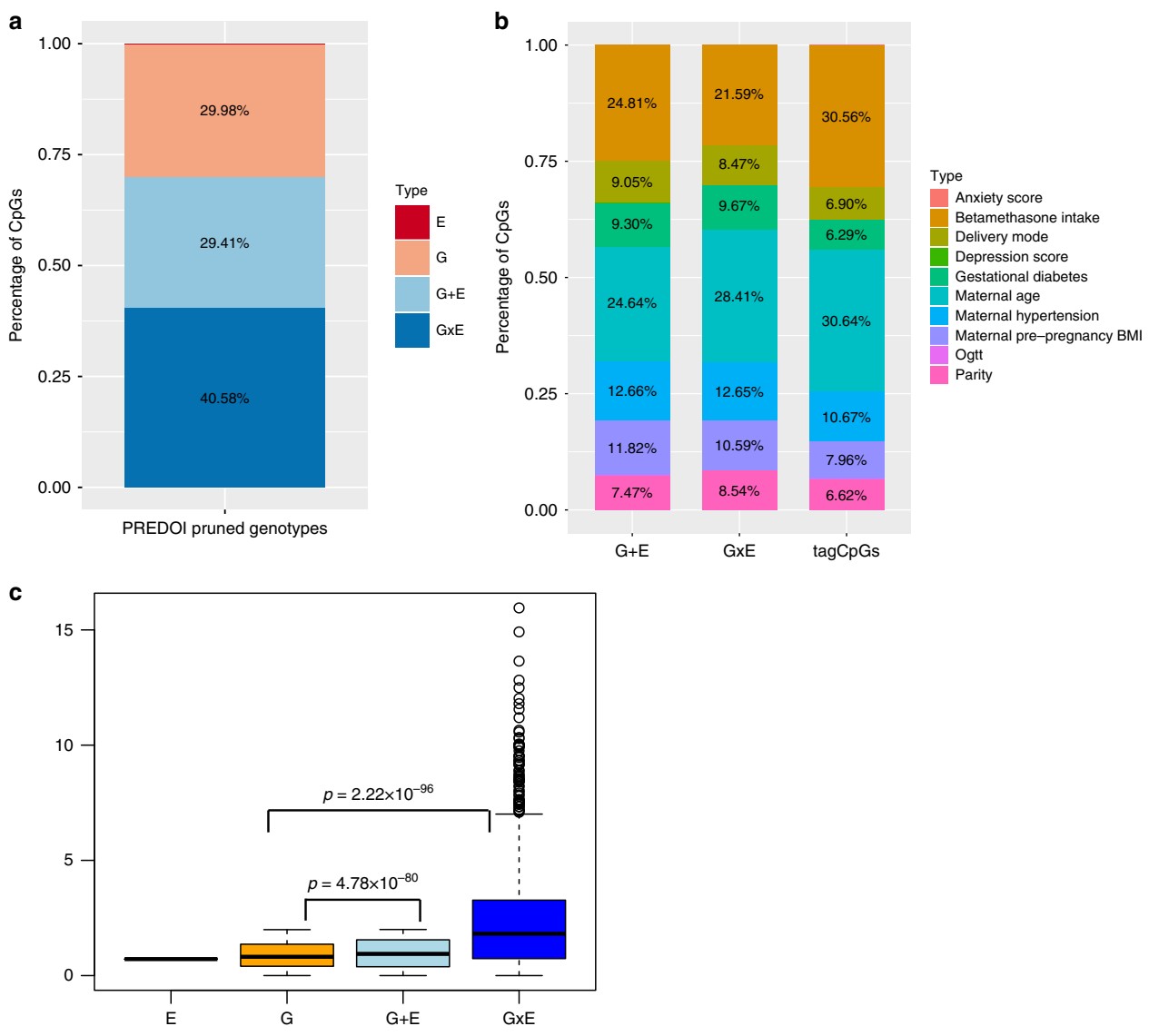

**Fig. 2** VMR analysis in pruned PREDO I dataset. **a** Percentage of models (G, E, GxE or G + E) with the lowest AIC explaining variable DNA methylation using the PREDO I dataset with pruned SNPs. **b** Distribution of the different types of prenatal environment included in the E model with the lowest AIC (right), in the combinations yielding the best model GxE (middle), or the best model G + E models (left). To increase readability all counts <3% have been omitted. **c** DeltaAIC, i.e, the difference in AIC, between best model and next best model, stratified by the best model. Y-axis denotes the delta AIC and the X-axis the different models. The median is depicted by a black line, the rectangle spans the first quartile to the third quartile, whiskers above and below the box show the location of minimum and maximum beta-values. P-values are based on Wilcoxon-tests

**Replication of best models in independent cohorts**. To assess whether the relative distribution of the best models for VMRs and DeepSEA variants was stable across different samples, we assessed the relative distribution of these models in 3 additional samples (DCHS I and DCHS II, UCI and PREDOII) with VMR data both from the Illumina 450 K as well as the IlluminaHumanEPIC arrays. Information on these cohorts is summarised in Table 1 and the number of VMRs, the distribution of VMR methylation levels, VMR length and specific SNP information are given in Supplementary Note 7 and Supplementary Fig. 6.

While major maternal factors overlapped among the cohorts - such as maternal age, delivery method, parity and depression during pregnancy - there were also differences, as the non-PREDO cohorts did not include betamethasone treatment but additionally included maternal smoking (see Table 1). Despites these differences and differences in the total number of VMRs, the overall pattern remained stable: in all 4 analyses, DCHS I,

DCHS II, UCI and PREDO II, we replicated that E alone models almost never explained most of the variances, while G alone models explained the most variance in up to 15% of the VMRs; G + E in up to 32%; and GxE models in up to 60% (see Fig. 5 and Table 2).

The importance of including G for a best model fit could also be observed for maternal smoking, described as one of the most highly replicated factors shaping the newborns' methylome[11] and present in the replication but not the discovery cohort PREDO I. These analyses are detailed in Supplementary Note 8.

We were also able to replicate our finding showing that GxE VMRs were enriched for OpenSea positions with a trend on the 450 K array (DCHS I, OR = 1.11, $p = 5.03 \times 10^{-02}$, Fisher-test) and significantly for the EPIC array data (PREDOII: $p = 2.96 \times 10^{-06}$, OR = 1.29, UCI: $p = 3.79 \times 10^{-02}$, OR = 1.09, DCHSII: $p = 2.91 \times 10^{-04}$, OR = 1.16, Fisher-tests). For all additional cohorts, the delta AIC for best model GxE to the next best

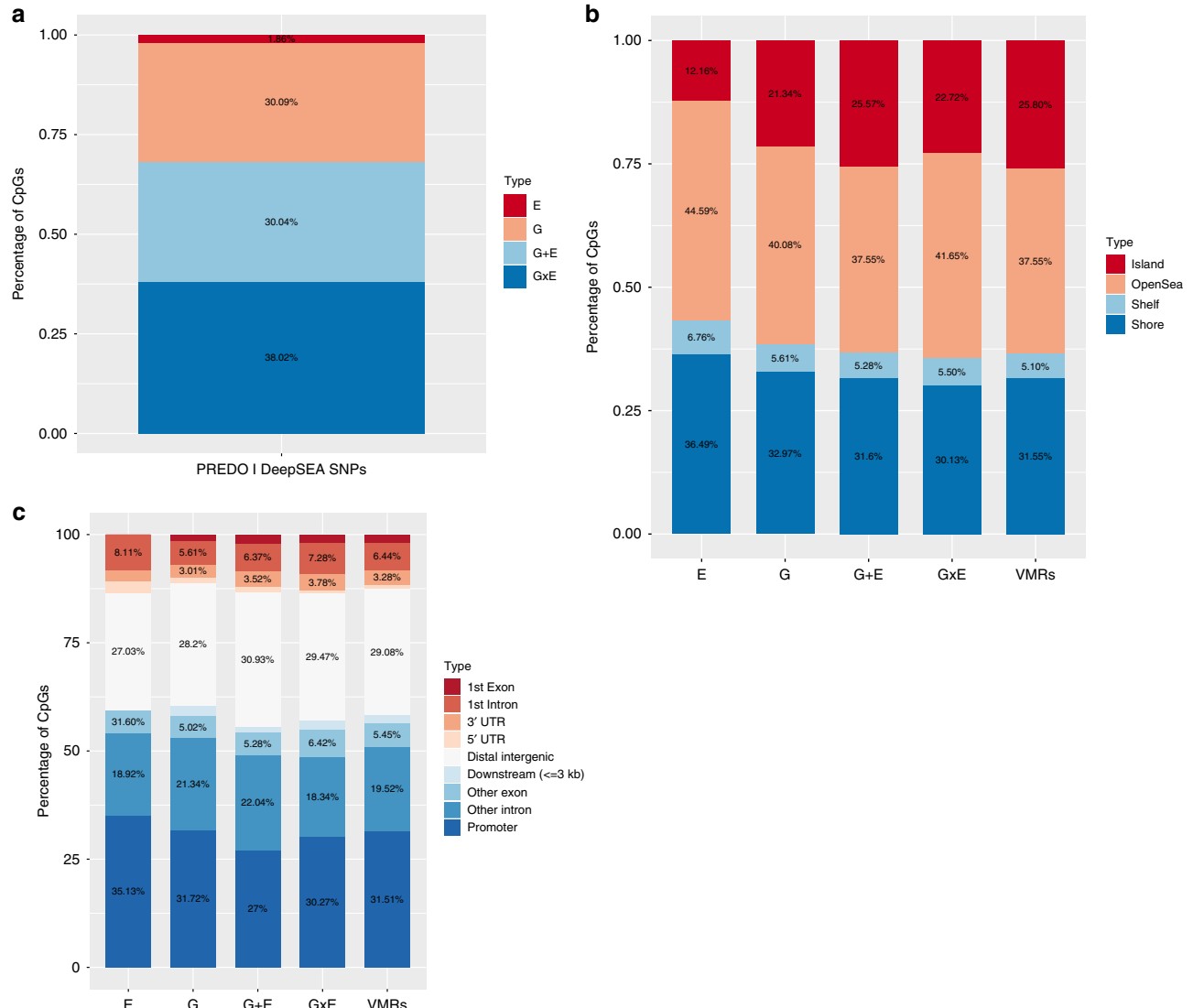

**Fig. 3** VMR analysis in DeepSEA annotated SNPs in PREDO I dataset. **a** Percentage of models (G, E, GxE or G + E) with the lowest AIC explaining variable DNA methylation using the PREDO I dataset with DeepSEA annotated SNPs. **b** Distribution of the locations of all VMRs and tagVMRs with best model E, G, G + E and GxE on the 450k array using only DeepSEA variants in relationship to CpG-Islands based on the Illumina 450 K annotation. **c** Distribution of gene-centric locations of all VMRs and tagVMRs with best model E, G, G + E and GxE on the 450k array using only DeepSEA variants

model was also significantly higher as compared to CpGs with G, E or G + E as the best model.

Overall, 387 tag CpGs overlapped between PREDO I, PREDO II, DCHS I and DCHS II (see Supplementary Fig. 7), which allowed us to test the consistency of the best models for specific VMRs across the different cohorts. Over 70% of the overlapping tagCPGs showed consistent best models in at least 3 cohorts (see Fig. 6) with GxE being the most consistent model (for over 60% of consistent models, see Supplementary Fig. 8). Focusing only on EPIC data (PREDO II, DCHSII and UCI), we identified more, namely 2091, tag CpGs that overlap across the three cohorts and here 86% show a consistent best model in at least two of the three cohorts, despite differences in study design, prenatal phenotypes and ethnicity.

Thus, the additional cohorts not only showed a consistent replication of the proportion of the models best explaining variance of VMRs but also consistency of the best model for specific VMRs. Within this context, we observed the GxE models were the most consistent models across the cohorts (see Supplementary Fig. 8), with 85% of the CpGs with consistent

models across 5 cohorts having GxE as the best model. Furthermore, we could validate specific GxE combinations between PREDO I and MoBa as shown as in the Supplementary Note 9, in Supplementary Data 17 and 18 and in Supplementary Fig. 9.

**Disease relevance**. Finally, we tested whether functional DeepSEA SNPs involved in only G, only GxE and only G + E models in PREDO I for their enrichment in GWAS hits. We used all functional SNPs and their LD proxies (defined as $r^2$ of at least 0.8 in the PREDO cohort and in maximal distance of 1MB to the target SNP) and performed enrichment analysis with the overlap of nominal significant GWAS hits. We selected for a broad spectrum of GWAS, including GWAS for complex disorders for which differences in prenatal environment are established as risk factors, but also including GWAS on other complex diseases. For psychiatric disorders, we used summary statistics of the Psychiatric Genomics Consortium (PGC) including association studies for autism[40], attention-deficit-hyperactivity disorder[41],

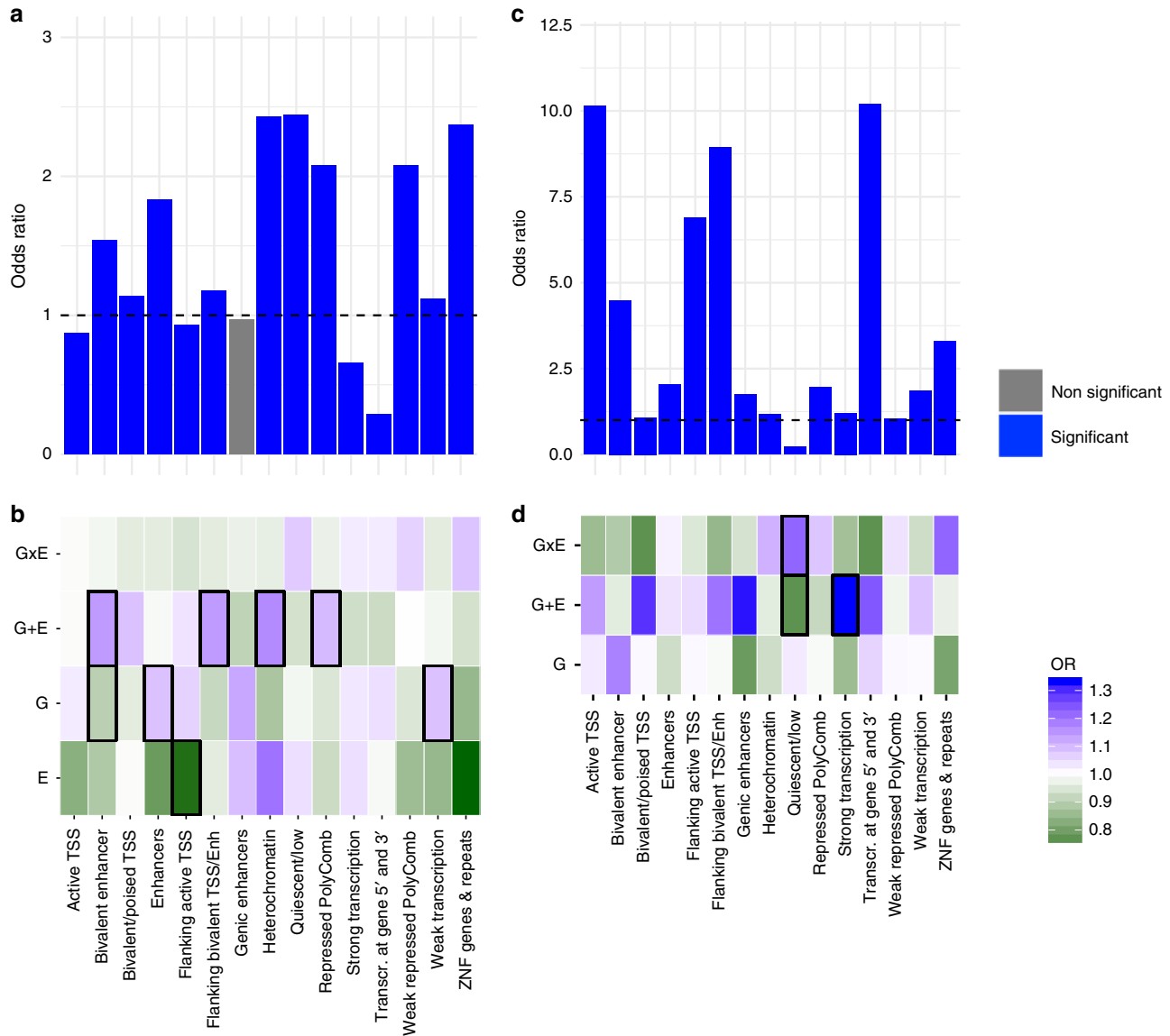

**Fig. 4** Functional annotation of VMR-mapping in DeepSEA annotated SNPs in PREDO I dataset. **a** Histone mark enrichment for all VMRs. The Y-axis denotes the fold enrichment/depletion as compared to no-VMRs. Blue bars indicate significant enrichment/depletion, grey bars non-significant differences based on Fisher-tests. **b** Histone mark enrichment for tagVMRs with best model E, G, G + E and GxE relative to all VMRs. Green colour indicates depletion, red colour indicates enrichment. Thick black lines around the rectangles indicate significant enrichment/depletion based on Fisher-tests. **c** Histone mark enrichment for all DeepSEA variants in the dataset. Blue bars indicate significant enrichment/depletion based on Fisher-tests. **d** Histone mark enrichment for all DeepSEA variants involved in models where either G, G + E or GxE is the best model as compared to all tested DeepSEA variants. Green colour indicates depletion, red colour indicates enrichment. Thick black lines around the rectangles indicate significant enrichment/depletion based on Fisher-tests

bipolar disorder[42], major depressive disorder[43], schizophrenia[44] and the cross-disorder associations including all five of these disorders[45]. Additionally, we included GWAS of inflammatory bowel disease[46], type 2 diabetes[47] and for BMI[48]. Nominal significant GWAS findings were enriched for DeepSEA variants and their LD proxies per se across psychiatric as well as non-psychiatric diseases (Fig. 7a). However, G, GxE and G + E DeepSEA variants showed a differential enrichment pattern above all DeepSEA variants (Fig. 7b), with strongest enrichments of GxE DeepSEA variants in GWAS of autism spectrum disorder ($p < 2.20 \times 10^{-16}$, OR = 2.07 above DeepSEA, Fisher-test), attention-deficit-hyperactivity disorder ($p < 2.20 \times 10^{-16}$, OR = 1.71, Fisher-test) and inflammatory bowel disease ($p < 2.20 \times 10^{-16}$, OR = 1.71, Fisher-test) and G + E DeepSEA variants in

GWAS for attention-deficit-hyperactivity disorder ($p = 9.54 \times 10^{-36}$, OR = 1.23, Fisher-test) and inflammatory bowel disease ($p = 1.85 \times 10^{-52}$, OR = 1.30, Fisher-test). While SNPs with strong main meQTL effects such as those within G and G + E models have been reported to be enriched in GWAS for common disease, we now also show this for SNPs within GxE models that often have non-significant main G effects.

## Discussion

We evaluated the effects of prenatal environmental factors and genotype on DNA methylation at VMRs identified in neonatal blood samples. We found that most variable methylation sites were best explained by either genotype and prenatal environment

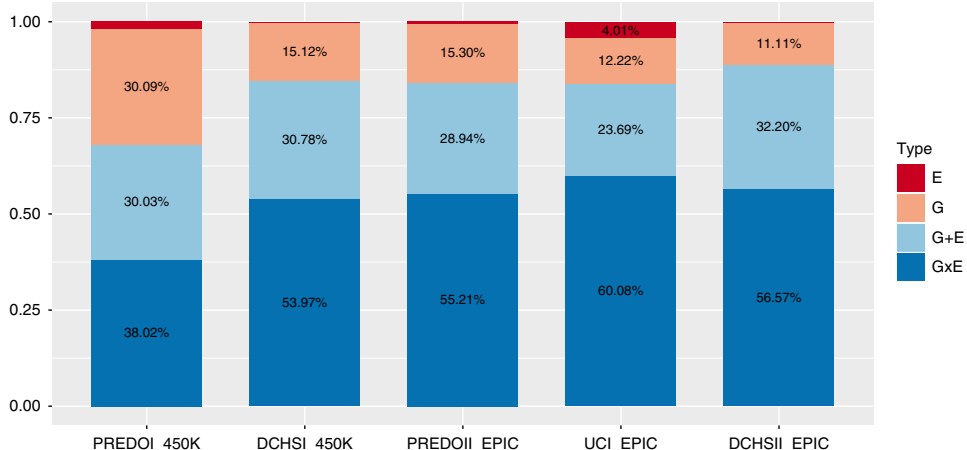

**Fig. 5** VMR analysis in PREDO I and replication datasets. Percentage of models (G, E, GxE or G + E) with the lowest AIC explaining variable DNA methylation in PREDO I (450 K), DCHS I (450 K), PREDO II (EPIC), UCI (EPIC) and DCHS II (EPIC)

**Table 2 VMRs and best models across cohorts**

| Cohort | PREDO I | PREDO II | DCHS I | DCHS II | UCI |
|---|---|---|---|---|---|
| Sample-size | 817 | 146 | 107 | 151 | 121 |
| Methylation array | Illumina 450 K | Illumina EPIC | Illumina 450 K | Illumina EPIC | Illumina EPIC |
| # VMRs | 3972 | 8547 | 6072 | 10,005 | 9525 |
| Proportion: best model E (%) | 2.0 | <1 | <1 | <1 | 4.1 |
| Best model G (%) | 30.0 | 15.0 | 15.8 | 11.5 | 12.8 |
| Best model G + E (%) | 30.0 | 29.0 | 29.8 | 32.1 | 24.1 |
| Best model GxE (%) | 38.0 | 56.0 | 54.3 | 56.3 | 59.0 |

interactions (GxE) or additive effects (G + E) of these factors, followed by main genotype effects. This pattern was replicated in independent cohorts and underscores the need to consider genotype in the study of environmental effects on DNA methylation.

In fact, VMRs best explained by G, G + E or GxE and their associated functional genetic variants were located in distinct genomic regions, suggesting that different combinatorial effects of G and E may impact VMRs with distinct downstream regulatory effects and thus possibly context-dependent impact on cellular function. We also observed that functional variants with best models G, G + E or GxE, all showed significant enrichment within GWAS signals for complex disorders beyond the enrichment of the functional variants themselves. While this was expected for G and G + E models based on results from previous studies[21,23,24,26], it was surprising for GxE SNPs, as these often do not have highly significant main genetic effects. Their specific enrichment in GWAS for common disorders supports the importance of these genetic variants that moderate environmental impact both at the level of DNA methylation but also, potentially, for disease risk.

The fact that GxE and G + E best explained the majority of VMRs (see Fig. 5) and that GxE models were selected by a larger margin than the other models (see Fig. 2c) was consistently found across all tested cohorts. These findings are in line with a previous report by Teh et al.[29] who performed a similar analysis based on AIC in umbilical cord tissue. Differences to the findings by Teh et al. are discussed in the Supplemental Discussion. Using data from four different cohorts, we not only saw comparable proportions of VMRs best explained by the different models, but also saw in the VMRs common across cohorts that specific VMRs had consistent best models (see Fig. 6). This is in line with the fact

that VMRs best explained by G, GxE or G + E show functional differences and may differentially impact gene regulation.

In addition to consistent findings using AIC-based approaches, we also observed some indication for validation of individual GxE and G + E combinations on selected VMRs using p-value based criteria, with a small number of specific G + E and GxE effects on VMRs replicating between the PREDO I and the MoBa cohort. The low number of specific replications could be due to lack of overall power as well as larger differences in prenatal factors between these two cohorts (see Table 1). As shown in Supplementary Fig. 4B, which specific G and E combinations best explain VMRs is also dependent on the specific prenatal factors. Larger and more homogenous cohorts regarding exposures will be needed for such analyses to be more conclusive.

While E alone was rarely the best model, it should be pointed out that main environmental effects on DNA methylation were observed (see Supplementary Data 3), and consistent with previous large meta-analyses such as in the case of maternal smoking (see Supplementary Note 7). Within the MoBa cohort, the cohort with the largest proportion of maternal smoking, 10% of all tagCpGs were best explained by maternal smoking alone. However, in all other cohorts, where smoking was less prevalent, the inclusion of genotypic effects in addition to maternal smoking explained more of the variance. This supports that while main E effects on the newborn methylome are present, genotype is an important factor that, in combination with E, may explain even more of the variance in DNA methylation.

VMRs best explained by either E, G, G + E or GxE and their associated functional SNPs were enriched for distinct genomics locations and chromatin states (see Fig. 4), suggesting that VMRs moderated by different combinations of G and E may in fact have

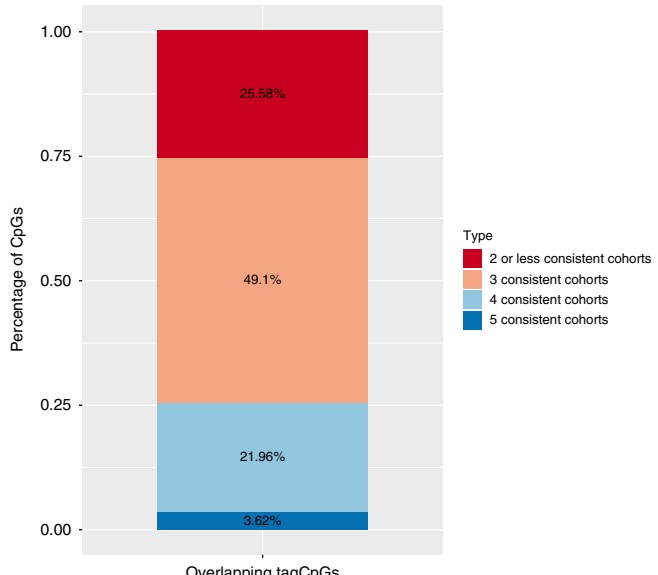

**Fig. 6** Consistency of best models across cohorts. Percentage of consistent best models in overlapping tag CpGs of PREDO I (450 K), DCHS I (450 K), PREDO II (EPIC), UCI (EPIC) and DCHS II (EPIC). Overlapping VMRs included significantly more CpGs as compared to all VMRs ($p < 2.2 \times 10^{-16}$, Wilcoxon-test, mean = 4.43)

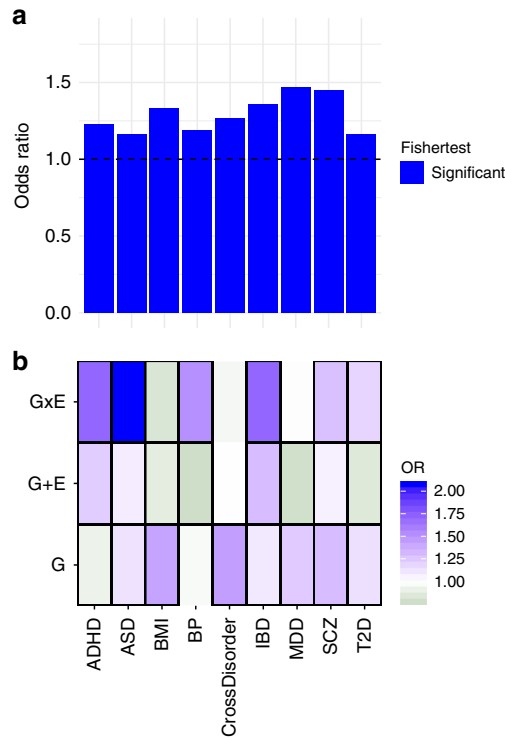

**Fig. 7** Enrichment of DeepSEA variants for GWAS associations. **a** Enrichment for nominal significant GWAS associations for all tested DeepSEA variants and their LD proxies for GWAS for ADHD (attention-deficit hyperactivity disorder), ASD (autism spectrum disorder), BMI (body mass index), BP (bipolar disorder), CrossDisorder, IBD (inflammatory bowel disease), MDD (major depressive disorder), SCZ (schizophrenia) and T2D (Type 2 diabetes). The Y-axis denotes the fold enrichment with regard to non-DeepSEAvariants. Blue bars indicate significant enrichment/ depletion based on Fisher-tests. **b** Enrichment for nominal significant GWAS hits for DeepSEA variants and their LD proxies involved in best models with G, G + E or GxE as compared to all tested DeepSEA variants. Green colour indicates depletion, red colour indicates enrichment. Thick black lines around the rectangles indicate significant enrichment/depletion based on Fisher-tests

distinct functional roles in gene regulation. Overall, VMRs best explained by GxE were consistently enriched for regions annotated to the OpenSea regions with lower CpG density and located farthest from CpG Islands[49]. Open Sea regions have been reported to be enriched for environmentally-associated CpGs with for example exposure to childhood trauma[50] and may harbour more long-range enhancers.

In addition to their position relative to CpG islands and their CpG content, G, GxE and G + E VMRs and their associated functional SNPs also showed distinct enrichments for chromatin marks. Compared to 450 K VMRs in general, VMRs with GxE as the best models were relatively depleted in regions surrounding the TSS, while VMRs with G + E were relatively enriched in these regions (see Fig. 4), suggesting that GxE VMRs are located at more distance from the TSS than G + E VMRs. To better map the potential functional variants in these models and to compare methylation-associated SNPs from a regulatory perspective, we used DeepSEA[38], a machine learning algorithm that predicts SNP functionality from the sequence context based on sequencing data for different regulatory elements in different cell lines using ENCODE data[39]. We identified the SNPs with putatively functional consequences on regulatory marks by DeepSEA and compared putative regulatory effects of G, G + E and GxE hits. Relative to the imputed non-DeepSEA SNPs contained in our dataset, these predicted functional DeepSEA SNPs were enriched for TSS and enhancer regions and depleted for quiescent regions, supporting their relevance in regulatory processes (see Fig. 4). Compared to DeepSEA SNPs overall, DeepSEA SNPs within the three different best models also showed distinct enrichment or depletion patterns. Similar to GxE VMRs, likely functional GxE SNPs also showed a relative depletion in TSS regions while G + E SNPs showed enrichment in genic enhancers. Overall, both the VMRs as well as the associated functional SNPs appear to be in distinct regulatory regions, depending on their best model. In addition, GxE functional SNP and tagCpGs were located farther apart than SNP/tagCpG pairs within G or G + E models (see Supplementary Fig. 5B), supporting a more long-range type of regulation in GxE interactions on molecular traits as compared to

all genes; a similar relationship has been reported previously for GxE with regard to gene expression in *C. elegans*[51,52].

SNPs associated with differences in gene expression but also DNA methylation have consistently been shown to be enriched among SNPs associated with common disorders in GWAS[21,24,26,53]. The functional genetic variants that were within G, GxE or G + E models predicting variable DNA methylation were even enriched in GWAS association results (beyond the baseline enrichment of DeepSea SNPs per se). The fact that such enrichment was observed for not only G and G + E SNPs, with strong main genetic effects, but also for GxE SNPs, with smaller to sometimes no main genetic effect on DNA methylation underscores the importance of also including SNPs within GxE models in the functional annotation of GWAS. A detailed catalogue of meQTLs that are responsive to environmental factors could support a better pathophysiological understanding of diseases for which risk is shaped by a combination of environment and genetic factors.

Finally, we want to note the limitations of this study. First, we restricted our analyses to specific DNA methylation array contents that are inherently biased as compared to genome-wide bisulfite sequencing, for example. In addition, we restricted our analysis to VMRs, which also limits the generalisability of the

findings, but also has advantages. Ong and Holbrooke[54] showed that this approach increases statistical power. Furthermore, VMRs are enriched for enhancers and transcription factor binding sites, overlap with GWAS hits[55] and are associated with gene expression of nearby genes at these sites[56]. VMRs in this study presented with intermediate methylation levels which have been shown to be enriched in regions of regulatory function, like enhancers, exons and DNase I hypersensitivity sites[57]. Hence, the effects of genotypes on DNA methylation levels in VMRs might be higher as compared to less variable CpG-sites. In addition, genotypes are measured with much less error as compared to environmental factors which may also reduce the overall explained variance in large cohorts.

Second, it has been reported that different cell types display different patterns of DNA methylation[55]. Therefore, the most variable CpG-sites may also include those that reflect differences in cord blood cell type proportions. To address this issue, all analyses were corrected for estimated cell proportions to the best of our current availability, so that differences in cell type proportion likely do not account for all of the observed effects. However, only replication in specific cell types will be able to truly assess the proportion of VMRs influenced by this.

Third, we used the AIC as main criterion for model fit[37] which is equivalent to a penalised likelihood-function. There are a variety of other model selection criteria[58] and choosing between these is an ongoing debate which also depends on the underlying research question. We decided to use the AIC as one of our main aims was to compare our results with the study of Teh et al.[29] in which this criterion was applied and as this method maybe more powerful for detecting GxE than for example model selection criteria based on lowest p-values.

Fourth, all reported interactions are statistical interactions and limited to a *cis* window around the CpG-site. Further experiments are required to assess whether these would also reflect biological/mechanistic interactions. Much larger cohorts will be needed to assess potential *trans* effects. Additional inclusion of further covariates such as maternal smoking or maternal age may further modify the effects of specific Es but is beyond the scope of this manuscript.

Fifth, as summarised in Table 1, results presented are based on cohorts which differ in ethnicity, assessed phenotypes, methylation and SNP arrays, processing pipelines and sample sizes. While all these factors may contribute to differences in the proportions of models across the cohorts, it also suggests that our findings are quite robust to these methodological issues.

Finally, our analyses are restricted to DNA methylation in neonatal blood and to pregnancy environments. Whether similar conclusions can be drawn for methylation levels assessed at a later developmental stage needs to be investigated.

We tested whether genotype, a combination of different prenatal environmental factors and the additive or the multiplicative interactive effects of both mainly influence VMRs in the newborn's epigenome. Our results show that G in combination with E are the best predictors of variance in DNA methylation. This highlights the importance of including both individual genetic differences as well as environmental phenotypes into epigenetic studies and also the importance of improving our ability to identify environmental associations. Our data also support the disease relevance of variants predicting DNA methylation together with the environment beyond main meQTL effects, and the view that there are functional differences of additive and interactive effects of genes and environment on DNA methylation. Improved understanding of these functional differences may also yield novel insights into pathophysiological mechanisms of common non-communicable diseases, as risk for all of these disorders is driven by both genetic and environmental factors.

## Methods

**The PREDO cohort.** The Prediction and Prevention of Preeclampsia and Intrauterine Growth Restriction (PREDO) Study is a longitudinal multicenter pregnancy cohort study of Finnish women and their singleton children born alive between 2006 and 2010 [30]. We recruited 1,079 pregnant women, of whom 969 had one or more and 110 had none of the known clinical risk factors for preeclampsia and intrauterine growth restriction. The recruitment took place when these women attended the first ultrasound screening at $12 + 0$–$13 + 6$ weeks $+$ days of gestation in one of the ten hospital maternity clinics participating in the study. The cohort profile[30] contains details of the study design and inclusion criteria.

**Ethics.** The study protocol was approved by the Ethical Committees of the Helsinki and Uusimaa Hospital District and by the participating hospitals. A written informed consent was obtained from all women.

**Maternal characteristics.** We tested 10 different maternal environments:

**Depressive symptoms.** Starting from $12 + 0$–$13 + 6$ gestational weeks $+$ days pregnant women filled in the 20 item Center for Epidemiological Studies Depression Scale (CES-D)[59] for depressive symptoms in the past 7 days. They filled in the CES-D scale biweekly until $38 + 0$–$39 + 6$ weeks $+$ days of gestation or delivery. We used the mean-value across all the CES-D measurements.

**Symptoms of anxiety.** At $12 + 0$–$13 + 6$ weeks $+$ days of gestation, women filled in the 20 item Spielberger's State Trait Anxiety Inventory (STAI)[60] for anxiety symptoms in the past 7 days. They filled in the STAI scale biweekly until $38 + 0$–$39 + 6$ weeks $+$ days of gestation or delivery. We used the mean-value across all these measurements.

**Betamethasone.** Antenatal betamethasone treatment (yes/no) was derived from the hospital records and the Finnish Medical Birth Register (MBR).

**Delivery method.** Mode of delivery (vaginal delivery vs. caesarean section) was derived from patient records and MBR.

**Parity.** Parity (number of previous pregnancies leading to childbirth) at the start of present pregnancy was derived from the hospital records and the MBR.

**Maternal age.** Maternal age at delivery (years) was derived from the hospital records and the MBR.

**Pre-pregnancy BMI.** Maternal pre-pregnancy BMI ($kg/m^2$), calculated from measurements weight and height verified at the first antenatal clinic visit at $8 + 4$ (SD $1 + 3$) gestational week was derived from the hospital records and the MBR.

**Hypertension.** Hypertension was defined as any hypertensive disorder including gestational hypertension, chronic hypertension and preeclampsia against normotension. Gestational hypertension was defined as systolic/diastolic blood pressure ≥140/90 mm Hg on ≥2 occasions at least 4 h apart in a woman who was normotensive before 20th week of gestation. Preeclampsia was defined as systolic/diastolic blood pressure ≥140/90 mm Hg on ≥2 occasions at least 4 h apart after 20th week of gestation and proteinuria ≥300 mg/24 h. Chronic hypertension was defined as systolic/diastolic blood pressure ≥140/90 mm Hg on ≥2 occasions at least 4 h apart before 20th gestational week or medication for hypertension before 20 weeks of gestation.

**Gestational diabetes and oral glucose tolerance test.** Gestational diabetes was defined as fasting, 1 h or 2 h plasma glucose during a 75 g oral glucose tolerance test ≥5.1, ≥10.0 and/or ≥8.5 mmol/L, respectively, that emerged or was first identified during pregnancy. We took the area under the curve from the three measurements as a single measure for the oral glucose tolerance test (OGTT) itself.

**Genotyping and imputation.** Genotyping was performed on Illumina Human Omni Express Exome Arrays containing 964,193 SNPs. Only markers with a call rate of at least 98%, a minor allele frequency of at least 1% and a p-value for deviation from Hardy-Weinberg-Equilibrium $>1.0 \times 10^{-06}$ were kept in the analysis. After QC, 587,290 SNPs were available.

In total, 996 cord blood samples were genotyped. Samples with a call rate below 98% ($n = 11$) were removed.

Any pair of samples with IBD estimates >0.125 was checked for relatedness. As we corrected for admixture in our analyses using MDS-components (see Supplementary Fig. 10), these samples were kept except for one pair which could not be resolved. From this pair we excluded one sample from further analysis. Individuals showing discrepancies between phenotypic and genotypic sex ($n = 1$)

were removed. We also checked for heterozygosity outliers but found none. Nine hundred and eighty-three participants were available in the final dataset.

Before imputation, AT and CG SNPs were removed. Imputation was performed using shapeit2 (http://mathgen.stats.ox.ac.uk/genetics_software/shapeit/shapeit.html) and impute2 (https://mathgen.stats.ox.ac.uk/impute/impute_v2.html). Chromosomal and base pair positions were updated to the 1000 Genomes Phase 3 reference set, allele strands were flipped where necessary.

After imputation, we reran quality control, filtering out SNPs with an info score <0.8, a minor allele frequency below 1% and a deviation from HWE with a $p$-value <$1.0 \times 10^{-06}$.

This resulted in a dataset of 9,402,991 SNPs. After conversion into best guessed genotypes using a probability threshold of 90%, we performed another round of QC (using SNP-call rate of least 98%, a MAF of at least 1% and a $p$-value threshold for HWE of $1.0 \times 10^{-06}$), after which 7,314,737 SNPs remained for the analysis.

For the evaluation of which model best explained the methylation sites, we pruned the dataset using a threshold of $r^2$ of 0.2 and a window-size of 50 SNPs with an overlap of 5 SNPs. The final, pruned dataset contained 788,156 SNPs. 36,241 of these variants were DeepSea variants (see Methods below).

**DNA methylation**. Cord blood samples were run on Illumina 450k Methylation arrays. The quality control pipeline was set up using the R-package *minfi*[61] (https://www.r-project.org). Three samples were excluded as they were outliers in the median intensities. Furthermore, 20 samples showed discordance between phenotypic sex and estimated sex and were excluded. Nine samples were contaminated with maternal DNA according to the method suggested by Morin et al.[62] and were also removed.

Methylation beta-values were normalised using the *funnorm* function[63]. After normalisation, two batches, i.e., slide and well, were significantly associated and were removed iteratively using the *Combat* function[64] in the *sva* package[65]. We excluded any probes on chromosome X or Y, probes containing SNPs and cross-hybridising probes according to Chen et al.[53] and Price et al.[66] Furthermore, any CpGs with a detection $p$-value >0.01 in at least 25% of the samples were excluded.

The final dataset contained 428,619 CpGs and 822 participants. For 817 of these, also genotypes were available.

An additional 161 cord blood samples were run on Illumina EPIC Methylation arrays.

Three samples were excluded as they were outliers in the median intensities. Three samples showed discordance between phenotypic sex and estimated sex and were excluded. Three samples were contaminated with maternal DNA and were also removed[62].

Methylation beta-values were normalised using the *funnorm* function[63] in the R–package *minfi*[61]. Three samples showed density artefacts after normalisation and were removed from further analysis. We excluded any probes on chromosome X or Y, probes containing SNPs and cross-hybridising probes according to Chen et al.[53], Price et al.[66] and McCartney et al.[67]. Furthermore, any CpGs with a detection $p$-value >0.01 in at least 25% of the samples were excluded. The final dataset contains 812,987 CpGs and 149 samples. After normalisation no significant batches were identified. For 146 of these samples, genotypic data was also available.

Cord blood cell counts were estimated for seven cell types (nucleated red blood cells, granulocytes, monocytes, natural killer cells, B cells, CD4(+)T cells, and CD8 (+)T cells) using the method of Bakulski et al.[68] which is incorporated in the R-package *minfi*[61].

**Identification of VMRs (variable methylated regions)**. The VMR approach was described by Ong and Holbrook[54]. We chose all 42,862 CpGs with a MAD score greater than the 90th percentile. For each CpG-site, the MAD (median absolute deviation) is defined as the median of the absolute deviations from each individual's methylation beta-value at this CpG-site to the CpG's median. A candidate VMR region was defined as at least two spatially contiguous probes which were at most 1 kb apart of each other. This resulted in 3982 VMRs in the 450 K samples and in 8547 VMRs in the EPIC sample. The CpG with the highest MAD scores was chosen as representative of the whole VMR in the statistical analysis.

**The Drakenstein cohort**. Details on this cohort and the assessed phenotypes can be found in refs. [34,35]. The birth cohort design recruits pregnant women attending one of two primary health care clinics in the Drakenstein sub-district of the Cape Winelands, Western Cape, South Africa – Mbekweni (serving a black African population) and TC Newman (serving a mixed ancestry population). Consenting mothers were enroled during pregnancy, and mother–child dyads are followed longitudinally until children reach at least 5 years of age. Mothers are asked to request that the father of the index pregnancy attend a single antenatal study visit where possible. Follow-up visits for mother–child dyads take place at the two primary health care clinics and at Paarl Hospital.

Pregnant women were eligible to participate if they were 18 years or older, were accessing one of the two primary health care clinics for antenatal care, had no intention to move out of the district within the following year, and provided signed written informed consent. Participants were enroled between 20 and 28 weeks' gestation, upon presenting for antenatal care visit. In addition, consenting fathers

of the index pregnancy when available were enroled in the study and attended a single antenatal study visit.

**Ethics**. The study was approved by the Faculty of Health Sciences, Human Research Ethics Committee, University of Cape Town (401/2009), by Stellenbosch University (N12/02/0002), and by the Western Cape Provincial Health Research committee (2011RP45). All participants provided written informed consent.

**Maternal characteristics**. After providing consent, participants were asked to complete a battery of self-report and clinician-administered measures at a number of antenatal and postnatal study visits. All assessed phenotypes are described in detail in ref. [34]. Here, we give a short outline on the phenotypes which were used in our analysis. Maternal parity was obtained from the antenatal record; maternal age was from the date of birth as recorded on the mothers' national identity document. The mode of delivery was ascertained by direct observation of the birth by a member of the study team as all births occurred at Paarl hospital. The SRQ-20[69] is a WHO-endorsed measure of psychological distress consisting of 20 items which assess non-psychotic symptoms, including symptoms of depressive and anxiety disorders. Each item is scored according to whether the participant responds in the affirmative (scored as 1) or negative (scored as 0) to the presence of a symptom. Individual items are summed to generate a total score. The Beck Depression Inventory (BDI-II) is a widely-used and reliable measure of depressive symptoms[70]. The BDI-II comprises 21 items, each of which assesses the severity of a symptom of major depression. Each item is assessed on a severity scale ranging from 0 (absence of symptoms) to 3 (severe, often with functional impairment). A total score is then obtained by summing individual item responses, with a higher score indicative of more severe depressive symptoms.

Smoking was assessed using The Alcohol, Smoking and Substance Involvement Screening Test (ASSIST)[71], a tool that was developed by the WHO to detect and manage substance use among people attending primary health care services. The tool assesses substance use and substance-related risk across 10 categories (tobacco, alcohol, cannabis, cocaine, amphetamine-type stimulants, inhalants, sedatives/sleeping pills, hallucinogens, opioids and other substances), as well as enquiring about a history of intravenous drug use. Total scores are obtained for each substance by summing individual item responses, with a higher score indicative of greater risk for substance-related health problems.

Hypertension was assessed by blood pressure measured antenatally.

**Genotyping and Imputation**. Genotyping in DCHS was performed using the Illumina PsychArray for those samples with 450k data, or the Illumina GSA for those samples with EPIC DNA methylation data (Illumina, San Diego, USA). For both array types, QC and imputation was the same; first, raw data was imported into Genome Studio and exported into R for QC. SNPs were filtered out if they had a tenth percentile GC score below 0.2 or an average GC score below 0.1, for a total of 140 SNPs removed. Phasing was performed using shapeit, and imputation was performed using impute2 with 1000 Genomes Phase 1 reference data. After imputation, we used qctool to filter out SNPs with an info score <0.8 or out of Hardy–Weinberg equilibrium. All SNPs with MAF <1% were removed.

As after imputation, only 5286 DeepSEA variants were available for those samples genotyped on the PsychArray and only 4049 for those samples genotyped on the GSAchip, we performed LD-pruning based on a threshold of $r^2$ of 0.2 and a window-size of 50 SNPs with an overlap of 5 SNPs. This resulted in 162,292 SNPs (PsychArray) and 176,553 SNPs (GSAchip).

**DNA methylation**. We performed basic quality control on data generated by either the 450k or EPIC arrays using Illumina's Genome Studio software for background subtraction and colour correction. Data was filtered to remove CpGs with high detection p values, those on the X or Y chromosome, or with previously identified poor performance. 450k data was normalised using SWAN and EPIC data using BMIQ, and both used ComBat to correct for chip (both), and row (450k only). Details for DNA methylation measurements and quality control have been published[62]. The final analysis was performed with 107 samples with methylation levels from the 450k array and 151 with methylation levels assessed on the EPIC array and available genotypes. Neonatal blood cell counts were estimated for seven cell types: nucleated red blood cells, granulocytes, monocytes, natural killer cells, B cells, CD4(+)T cells, and CD8(+)T cells[68].

**VMRs**. We identified 6072 candidate VMRs in DCHS I and 10,005 candidate VMRs in DCHS II.

**The UCI cohort**. Mothers and children were part of an ongoing, longitudinal study, conducted at the University of California, Irvine (UCI), for which mothers were recruited during the first trimester of pregnancy[31–33]. All women had singleton, intrauterine pregnancies. Women were not eligible for study participation if they met the following criteria: corticosteroids, or illicit drugs during pregnancy (verified by urinary cotinine and drug toxicology). Exclusion criteria for the newborn were preterm birth (i.e., less than 34 weeks of gestational age at birth), as well as any congenital, genetic, or neurologic disorders at birth.

**Ethics**. The UCI institutional review board approved all study procedures and all participants provided written informed consent.

**Maternal characteristics**. Maternal sociodemographic characteristics (age, parity) were obtained via a standardised structured interview at the first pregnancy visit. Maternal pre-pregnancy BMI (weight kg/height m$^2$) was computed based on pre-pregnancy weight abstracted from the medical record, and maternal height was measured at the research laboratory during the first pregnancy visit. Obstetric risk conditions during pregnancy, including presence of gestational diabetes and hypertension, and delivery mode were abstracted from the medical record. At each pregnancy visit the Center for Epidemiological Studies Depression Scale[59] and the State scale from the State–Trait Anxiety Inventory[60] were administered. For individuals with <3 missing items on any scale at any time point, the mean responses for that scale were calculated and then multiplied by the total number of items in the respective scale, to generate total scale scores that are comparable to those generated from participants without any missing data. We used the average depression and anxiety score throughout pregnancy in the calculations. Maternal smoking during pregnancy was determined by maternal self-report and verified by measurement of urinary cotinine concentration. Urinary cotinine was assayed in maternal samples collected at each trimester using the Nicotine/COT(Cotinine)/ Tobacco Drug Test Urine Cassette (http://www.meditests.com/nicuintescas.html), which involves transferring 4 drops of room temperature urine into the well of the cassette, and employs a cutoff for COT presence of 200 ng/ml. Endorsement of smoking or detection of urinary COT in any trimester was coded as 1, and absence of evidence for smoking in any trimester coded as 0.

**Genotyping**. Genomic DNA was extracted from heel prick blood samples and used for all genomic analysis. Genotyping was performed on Illumina Human Omni Express (24 v1.1) Arrays containing 713,014 SNPs. All samples had a high call rate (above 97%). SNPs with a minor allele frequency >5% and a p-value for deviation from Hardy-Weinberg-Equilibrium $>1.0 \times 10^{-25}$ were retained for analysis. After QC, 602,807 SNPs were available.

**Imputation**. Before imputation, chromosomal and base pair positions were updated to the Haplotype Reference Consortium (r1.1) reference set, allele strands were flipped where necessary. Phasing was performed using EAGLE2 (https://data. broadinstitute.org/alkesgroup/Eagle/) and imputation was performed using PBWT (https://github.com/VertebrateResequencing/pbwt). Imputed SNPs with an info score <0.8, duplicates and ambiguous SNPs were removed resulting in 21,341,980 SNPs. All SNPs with MAF <0.01 were removed. Of the remaining SNPs, 19,530 were DeepSEA variants.

**DNA methylation**. DNAm analysis using the Infinium Illumina MethylationEPIC BeadChip (Illumina, Inc., San Diego, CA) was performed according to the manufacturer´s guidelines in using genomic DNA derived from neonatal heel prick samples. Quality Control carried out in *minfi*[61]. No outliers were detected in the median intensities of methylated and unmethylated channels. All samples had a high call rate of at least 95% and their predicted sex was the same as the phenotypic sex. We removed CpGs with a high detection value ($p < 0.0001$), probes missing >3 beads in >5% of the cohort, in addition to non-specific/cross-hybridising and SNP probes[66,67]. Methylation beta-values were normalised using functional normalisation (*funnorm*)[63]. We also iteratively adjusted the data for relevant technical factors, i.e., array row, experimental batch and sample plate, using *Combat*[64]. The final dataset contained 768,910 CpGs. Neonatal blood cell counts were estimated for seven cell types: nucleated red blood cells, granulocytes, monocytes, natural killer cells, B cells, CD4(+)T cells, and CD8(+)T cells[68]. The final dataset contained 121 samples with available genotypes and methylation values.

**VMRs**. Applying the same procedure as for PREDO I and PREDO II, we identified 9525 candidate VMRs in the ICU cohort.

**The MoBa cohort**. Participants represent two subsets of mother-offspring pairs from the national Norwegian Mother and Child Cohort Study (MoBa)[72]. MoBa is a prospective population-based pregnancy cohort study conducted by the Norwegian Institute of Public Health. The years of birth for MoBa participants ranged from 1999 to 2009. MoBa mothers provided written informed consent. Each subset is referred to here as MoBa1 and MoBa2. MoBa1 is a subset of a larger study within MoBa that included a cohort random sample and cases of asthma at age 3 years[73]. We previously reported an association between maternal smoking during pregnancy and differential DNA methylation in MoBa1 newborns[74]. We subsequently measured DNA methylation in additional newborns (MoBa2) in the same laboratory (Illumina, San Diego, CA)[11]. MoBa2 included cohort random sample plus cases of asthma at age 7 years and non-asthmatic controls. Years of birth were 2002–2004 for children in MoBa1, 2000–2005 for MoBa2.

**Ethics**. The establishment and data collection in MoBa obtained a license from the Norwegian Data Inspectorate and approval from The Regional Committee for Medical Research Ethics. Both studies were approved by the Regional Committee

for Ethics in Medical Research, Norway. In addition, MoBa1 and MoBa2 were approved by the Institutional Review Board of the National Institute of Environmental Health Sciences, USA.

**Maternal characteristics**. To replicate specific GxE and G + E from PREDO I, we focused on those characteristics which were available in both cohorts: maternal age, pre-pregnancy BMI and hypertension.

Within MoBa, the questionnaires at weeks 17 and 30 include general background information as well as details on previous and present health problems and exposures. The birth record from the Medical Birth Registry of Norway[75] which includes maternal health during pregnancy as well as procedures around birth and pregnancy outcomes, is integrated in the MoBa database.

**Genotyping and imputation**. DNA was extracted from the MoBa biobank and genotyped on the Illumina HumanExomeCore platform. The genotypes were called with GenomeStudio software. Phasing and imputation were done using shapeit2 (https://mathgen.stats.ox.ac.uk/genetics_software/shapeit/shapeit.html) and impute2 (https://mathgen.stats.ox.ac.uk/impute/impute_v2.html) with the thousand genomes phase 3 reference panel for the European population. Variants with a imputation score of <0.8 and with a minor allele frequency below 1% were filtered out.

**DNA methylation**. Details of the DNA methylation measurements and quality control for the MoBa1 participants were previously described[36] and the same protocol was implemented for the MoBa2 participants. Briefly, at birth, umbilical cord blood samples were collected and frozen at birth at −80 °C. All biological material was obtained from the Biobank of the MoBa study[36]. Bisulfite conversion was performed using the EZ-96 DNA Methylation kit (Zymo Research Corporation, Irvine, CA) and DNA methylation was measured at 485,577 CpGs in cord blood using Illumina's Infinium HumanMethylation450 BeadChip[76]. Raw intensity (.idat) files were handled in R using the *minfi* package to calculate the methylation level at each CpG as the beta-value ($\beta$ = intensity of the methylated allele (M)/ (intensity of the unmethylated allele (U)+ intensity of the methylated allele (M) + 100)) and the data was exported for quality control and processing. Control probes ($N = 65$) and probes on X ($N = 11\ 230$) and Y ($N = 416$) chromosomes were excluded in both datasets. Remaining CpGs missing >10% of methylation data were also removed ($N = 20$ in MoBa1, none in MoBa2). Samples indicated by Illumina to have failed or have an average detection $p$ value across all probes <0.05 ($N = 49$ MoBa1, $N = 35$ MoBa2) and samples with gender mismatch ($N = 13$ MoBa1, $N = 8$ MoBa2) were also removed. For MoBa1 and MoBa2, we accounted for the two different probe designs by applying the intra-array normalisation strategy Beta Mixture Quantile dilation (BMIQ)[77]. The Empirical Bayes method via *ComBat* was applied separately in MoBa1 and MoBa2 for batch correction using the *sva* package in R[65]. After quality control exclusions, the sample sizes were 1068 for MoBa1 and 685 for MoBa2.

After QC, the total number of samples was 1732, with 1592 overlapping with the methylation samples. Specific G + E and GxE associations were calculated in the combined dataset of MoBa1 and MoBa2, while VMR analysis was conducted in MoBa1 only.

**Regression analysis**. Linear regression analysis was conducted using the *lm* function in R 3.3.1 (https://www.r-project.org). We included the child's sex, gestational age, seven estimated cell counts as well as the first two (PREDO I and PREDO II), first three (UCI) and first five (DCHS I and II) principal components of the MDS analysis on the genotypes in the model. The corresponding plot of the first ten MDS-components in PREDO is depicted in Figure S4. SNP genotypes were recoded into a count of 0, 1 or 2 representing the number of minor allele copies. For each VMR site, we tested SNPs located in a 1MB window up- and downstream of the specific site. In PREDO and UCI, we restricted the analysis to DeepSEA variants while we used the pruned SNP-set in DCHS.

For each VMR, we tested four models:

(1) Methylation at tagCpG ~ covariates + environment
(2) Methylation at tagCpG ~ covariates + SNP
(3) Methylation at tagCpG ~ covariates + SNP + environment
(4) Methylation at tagCpG ~ covariates + SNP + environment + SNP×environment

In model (1) we included all ten different environments, in model (2) all DeepSEA cis SNPs and in models (3) and (4) all possible environment-cis-SNP combinations. Please also see Fig. 1.

For each model, the AIC, Akaike's information criterion[37] was calculated and the model with the lowest AIC was chosen as the best model. The AIC was obtained using the *AIC* function in R 3.3.1 (https://www.r-project.org).

*P*-values were obtained from the summary function and adjusted for the number of tested Es (E model), of tested cis SNPs (G model) or of tested cis SNP-environment combinations (G + E/GxE model) using Bonferroni-correction. Afterwards, we used FDR to correct for all tested tagCpGs (all models) using *p. adjust* in R.

**Enrichment analyses**. With regard to enrichment for VMRs, CpG-site within VMRs were compared to all other CpG-sites on the 450 K array located in non-VMR-regions. With regard to enrichment for VMRs best explained by G, G + E or GxE, tagCpGs best explained by the specific model were compared to tagCpGs best explained by any of the other models. For enrichment tests for DeepSEA SNPs, non-DeepSEA SNPs present in our dataset were used as comparison group. Enrichment tests were performed based on a hyper-geometric test, i.e. a Fisher-test. The significance levels was set at $p < 0.05$.

With regard to enrichment for GWAS hits, DeepSEA variants were matched to GWAs variants based on chromosome and position (hg19). To check for enrichment for nominal significant GWAS hits, the full summary statistics were derived from the respective publication.

Histone ChiP-seq peaks from Roadmap Epigenomics project for blood and embryonic stem cells were downloaded from http://egg2.wustl.edu/roadmap/data/byFileType/peaks/consolidated/broadPeak/.

The pre-processed consolidated broad peaks from the uniform processing pipeline of the Roadmap project were used.

**Genomic annotation mapping**. CpG sites were mapped to the genome location according to Illumina's annotation using the R-package *minfi*.

**DeepSEA analysis**. Pretrained DeepSEA model was downloaded from: http://deepsea.princeton.edu/media/code/deepsea.v0.94.tar.gz and variant files in VCF format are used for producing e-values. VCF files were first split into smaller files each containing one million variants and the model was run using the command line on a server with a NVIDIA Titan X GPU card.

We reran our models using only DeepSEA variants which had been identified by the algorithm of Zhou and Troyanskaya[38]. This method predicts functionality of a SNP based on the DNA-sequence. We included all 212,210 variants with a functional significance e-value below $5 \times 10^{-05}$. The e-values represent the significance of the regulatory impact of given variants compared to one million random variants.

**Random-effects meta-analysis**. GxE and G + E result for PREDO and for MoBa were meta-analysed using a random-effects model in the R-package rmeta. Replication was defined as DeepSEA-tagCpG-environment combinations showing the same effect direction in both cohorts, presenting with smaller $p$-values as for PREDO alone and with a FDR-corrected $p$-value (across all combinations tested in the meta-analysis) below 0.05.

**Reporting summary**. Further information on research design is available in the Nature Research Reporting Summary linked to this article.

## Data availability

Due to ethical issues and consent the datasets analysed during the current study are not publicly available. However, an interested researcher can obtain a de-identified dataset after approval from the PREDO Study Board. Data requests may be subject to further review by the national register authority and by the ethical committees. Data can be obtained upon reasonable request from the PREDO Study Board (predo.study@helsinki.fi) or individual researchers. The summary statistics of the best models for PREDO I are accessible at: https://doi.org/10.6084/m9.figshare.8074964.

For access to the UCI cohort, please contact claudia.buss@charite.de, for access to DCHS please contact Heather.Zar@uct.ac.za, for MoBa access please apply for data access at https://www.fih.no

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

## Acknowledgements

We want to thank Susanne Sauer and Maik Ködel for their technical assistance and Jessica Keverne for language editing. We thank all mothers who took part in the on-going PREDO study. We are grateful to all the families in Norway who participate in the on-going MoBa cohort study. We thank the Drakenstein Child Health Study staff, and the clinical and administrative staff of the Western Cape Government Department of Health at Paarl Hospital and at the clinics for support of the Study. We also thank our collaborators and students. Finally, we thank all mothers and children enroled in the Drakenstein Child Health Study. We thank the research participants and employees of 23andMe, Inc. for their contribution to this study. This work was supported by the Academy of Finland (E.K., H.L., K.R., and J.L.); University of Helsinki Research Funds (J.L., M.L.P., and H.L.), British Heart Foundation (RMR); Tommy's (RMR); European Commission (EK, KR, Horizon 2020 Award SC1–2016-RTD-733280 RECAP); NorFace DIAL (E.K., KR PremLife); Foundation for Pediatric Research (E.K.); Juho Vainio Foundation (E.K.); Novo Nordisk Foundation (E.K.); Signe and Ane Gyllenberg Foundation (E.K., K.R.); Sigrid Jusélius Foundation (E.K.); Finnish Medical Foundation (H.L.); Jane and Aatos Erkko Foundation (H.L.); Päivikki and Sakari Sohlberg Foundation (H.L., P.M.V.); the Clinical Graduate school in Pediatrics and Obstetrics/Gynaecology at University of Helsinki (P.M.V.). The Norwegian Mother and Child Cohort Study is supported by the Norwegian Ministry of Health and Care Services and the Ministry of Education and Research, NIH/NIEHS (contract no N01-ES-75558), NIH/NINDS (grant no.1 UO1 NS 047537–01 and grant no.2 UO1 NS 047537–06A1). For this work, MoBa 1 and 2 were supported by the Intramural Research Program of the NIH, National Institute of Environmental Health Sciences (Z01-ES-49019) and the Norwegian Research Council/ BIOBANK (grant no 221097). This work was also partly supported by the Research Council of Norway through its Centres of Excellence funding scheme, project number 262700. The Drakenstein Child Health Study is supported by the Bill and Melinda Gates Foundation (OPP 1017641); with additional support for this work from the Eunice Kennedy Shriver National Institute of Child Health and Human Development of the National Institutes of Health (NICHD) under Award Number R21HD085849; and the Fogarty International Center (FIC). The content is solely the responsibility of the authors and does not necessarily represent the official views of the National Institutes of Health. Additional support for H.J.Z., D.J.S. and N.K., and for research reported in this publication was by the South African Medical Research Council (SAMRC); N.K. receives support from the SAMRC under a Self-Initiated Research Grant. The views and opinions

expressed are those of the authors and do not necessarily represent the official views of the SAMRC. This work was also funded by the German Federal Ministry of Education and Research through the Research Consortium Integrated Network IntegraMent (grant 01ZX1314H) under the auspices of the e:Med Programme (NSM). The UCI cohort was supported by a European Research Area Network (ERA Net) Neuron grant (01EW1407A, CB) and National Institutes of Health grant (R01 HD-060628, CB) as well as NIH grant R01 MH-105538 (PDW). This work was also funded by the Canadian Institute for Advanced Research, Child and Brain Development Program, Toronto, ON, Canada (KJOD).

## Author contributions

D.C. and E.B.B. conceived the analyses. J.L., M.L.P., E.H., E.K., H.L., P.M.V., R.M.R. and K.R. conceptualised and planned the PREDO study and collected the data. C.M.P., W.N., S.H. and S.J.L. conceptualised and planned the MoBa study and collected the data. C.B., S.E., P.D.W., and K.J.O.D. conceptualised and planned the UCI study and collected the data. D.T.S.L., J.L.M. and E.G. performed the DNA methylation and genotyping arrays for the UCI and DCH studies. D.J.S., N.K., and H.J.Z. designed and undertook the DCHS; M.J.M., M.S.K., and K.C.K. were involved in testing and analysis of epigenetic data; S.D. was involved in testing and analysis of genetic data. Major Depressive Disorder Working Group of the Psychiatric Genomics Consortium calculated summary statistics for enrichment tests. D.C., G.E., C.M.P. and M.J.J. ran the statistical analysis. N.S.M., I.K. and F.J.T. co-supervised statistical analysis. D.C. and E.B.B. wrote the manuscript with contributions from G.E., S.J.L., C.M.P., K.R., J.L.; D.C., J.L., K.R. and E.B.B. interpreted the results. All authors contributed to and approved the final version of the manuscript.

## Additional information

**Competing interests:** E.B.B. is co-inventor on the following patent applications: FKBP5: a novel target for antidepressant therapy. European Patent# EP 1687443 B1; Polymorphisms in ABCB1 associated with a lack of clinical response to medicaments.

United States Patent # 8030033; Means and methods for diagnosing predisposition for treatment emergent suicidal ideation (TESI). European application number: 08016477.5 International application number: PCT/EP2009/061575. The remaining authors declare no competing interests.

Darina Czamara [1], Gökçen Eraslan [2,3], Christian M. Page [4,5], Jari Lahti [6,7], Marius Lahti-Pulkkinen [6,8], Esa Hämäläinen[9], Eero Kajantie[10,11,12], Hannele Laivuori[13,14,15,16], Pia M. Villa[13], Rebecca M. Reynolds[8], Wenche Nystad[17], Siri E. Håberg[5], Stephanie J. London [18], Kieran J. O'Donnell[19,20], Elika Garg [19], Michael J. Meaney[19,20,21], Sonja Entringer[22,23], Pathik D. Wadhwa[23,24], Claudia Buss [22,23], Meaghan J. Jones[25], David T.S. Lin [25], Julie L. MacIsaac[25], Michael S. Kobor[25], Nastassja Koen[26,27], Heather J. Zar[28], Karestan C. Koenen[29], Shareefa Dalvie[26], Dan J. Stein[26,27], Ivan Kondofersky[2,30], Nikola S. Müller[2], Fabian J. Theis [2,30], Major Depressive Disorder Working Group of the Psychiatric Genomics Consortium[†], Katri Räikkönen[6] & Elisabeth B. Binder[1,31]

[1]Max-Planck-Institute of Psychiatry, Department of Translational Research in Psychiatry, Munich 80804, Germany. [2]Institute of Computational Biology, Helmholtz-Zentrum München, German Research Center for Environmental Health, Neuherberg 85764, Germany. [3]School of Life Sciences, Weihenstephan, Technische Universität München, Freising 85354, Germany. [4]Oslo Centre for Biostatistics and Epidemiology, Research Support Unit, Oslo University Hospital, Oslo 0372, Norway. [5]Center for Fertility and Health, Norwegian Institute of Public Health, Oslo 0213, Norway. [6]Department of Psychology and Logopedics, Faculty of Medicine, University of Helsinki, Helsinki 00014, Finland. [7]Helsinki Collegium for Advanced Studies, University of Helsinki, Helsinki 00101, Finland. [8]British Heart Foundation Centre for Cardiovascular Science, Queen's Medical Research Institute, University of Edinburgh, Edinburgh EH16 4TJ, UK. [9]HUSLAB and Department of Clinical Chemistry, Helsinki University, Helsinki 00290, Finland. [10]Oulu University Hospital and University of Oulu, PEDEGO Research Unit, MRC Oulu 90014, Finland. [11]Hospital for Children and Adolescents, University of Helsinki and Helsinki University Hospital, Helsinki 00029, Finland. [12]National Institute for Health and Welfare, Helsinki 00271, Finland. [13]Medical and Clinical Genetics and Obstetrics and Gynaecology University of Helsinki and Helsinki University Central Hospital, Helsinki 00014, Finland. [14]Institute for Molecular Medicine Finland, Helsinki Institute of Life Science, University of Helsinki, Helsinki 00014, Finland. [15]Faculty of Medicine and Life Sciences, University of Tampere, Tampere 33100, Finland. [16]Department of Obstetrics and Gynecology, Tampere University Hospital, Tampere 33100, Finland. [17]Department of Chronic Diseases and Ageing, Norwegian Institute of Public Health, Oslo 0213, Norway. [18]Epidemiology Branch, National Institute of Environmental Health Sciences, National Institutes of Health, U.S. Department of Health and Human Services, Research Triangle Park, North Carolina 20814, USA. [19]Ludmer Centre for Neuroinformatics and Mental Health, Douglas Mental Health University Institute, McGill University, Montreal H3A 2B4 QC, Canada. [20]Sackler Program for Epigenetics and Psychobiology at McGill University, Montreal H3A 0G4 QC, Canada. [21]Singapore Institute for Clinical Sciences, Singapore 117609, Singapore. [22]Charité – Universitätsmedizin Berlin, corporate member of Freie Universität Berlin, Humboldt-Universität zu Berlin, and Berlin Institute of Health (BIH), Institute of Medical Psychology, Berlin 10117, Germany. [23]University of California, Irvine, Development, Health, and Disease Research Program, Orange, CA 92697, USA. [24]Department of Psychiatry and Human Behavior, Obstetrics and Gynecology, and Epidemiology, University of California, Irvine, School of Medicine, Irvine, CA 92697, USA. [25]Centre for Molecular Medicine and Therapeutics, Department of Medical Genetics, University of British Columbia and the BC Children's Hospital Research Institute, Vancouver V5Z 4H4 BC, Canada. [26]Department of Psychiatry and Mental

Health, University of Cape Town, Cape Town 7925, South Africa. [27]South African Medical Research Council (SAMRC), Unit on Risk and Resilience in Mental Disorders, Cape Town 7505, South Africa. [28]Department of Paediatrics & Child Health and SAMRC Unit on Child and Adolescent Health, University of Cape Town, Cape Town 7505, South Africa. [29]Department of Epidemiology, Harvard T. H. Chan School of Public Health, Boston, MA 02115, USA. [30]Department of Mathematics, Technische Universität München, Munich 85748, Germany. [31]Department of Psychiatry and Behavioral Sciences, Emory University School of Medicine, Atlanta 30329, USA. [†]A full list of consortium members appears at the end of the paper.

## Major Depressive Disorder Working Group of the Psychiatric Genomics Consortium

Naomi R. Wray[32,33], Stephan Ripke[34,35,36], Manuel Mattheisen[37,38,39,40], Maciej Trzaskowski[32], Enda M. Byrne[32], Abdel Abdellaoui[41], Mark J. Adams[42], Esben Agerbo[40,43,44], Tracy M. Air[45], Till F.M. Andlauer[1,46], Silviu-Alin Bacanu[47], Marie Bækvad-Hansen[40,48], Aartjan T.F. Beekman[49], Tim B. Bigdeli[47,50], Douglas H.R. Blackwood[42], Julien Bryois[51], Henriette N. Buttenschøn[39,40,52], Jonas Bybjerg-Grauholm[40,48], Na Cai[53,54], Enrique Castelao[55], Jane Hvarregaard Christensen[38,39,40], Toni-Kim Clarke[42], Jonathan R.I. Coleman[56], Lucía Colodro-Conde[57], Baptiste Couvy-Duchesne[58,59], Nick Craddock[60], Gregory E. Crawford[61,62], Gail Davies[63], Ian J. Deary[63], Franziska Degenhardt[64,65], Eske M. Derks[57], Nese Direk[66,67], Conor V. Dolan[41], Erin C. Dunn[68,69,70], Thalia C. Eley[56], Valentina Escott-Price[71], Farnush Farhadi Hassan Kiadeh[72], Hilary K. Finucane[58,73], Andreas J. Forstner[64,65,74,75], Josef Frank[76], Héléna A. Gaspar[56], Michael Gill[77], Fernando S. Goes[78], Scott D. Gordon[79], Jakob Grove[38,39,40,80], Lynsey S. Hall[42,81], Christine Søholm Hansen[40,48], Thomas F. Hansen[82,83,84], Stefan Herms[64,65,75], Ian B. Hickie[85], Per Hoffmann[47,64,65], Georg Homuth[86], Carsten Horn[87], Jouke-Jan Hottenga[41], David M. Hougaard[40,48], Marcus Ising[88], Rick Jansen[49], Eric Jorgenson[89], James A. Knowles[90], Isaac S. Kohane[91,92,93], Julia Kraft[35], Warren W. Kretzschmar[94], Jesper Krogh[95], Zoltán Kutalik[96,97], Yihan Li[94], Penelope A. Lind[57], Donald J. MacIntyre[98,99], Dean F. MacKinnon[78], Robert M. Maier[33], Wolfgang Maier[100], Jonathan Marchini[101], Hamdi Mbarek[41], Patrick McGrath[102], Peter McGuffin[56], Sarah E. Medland[57], Divya Mehta[33,103], Christel M. Middeldorp[41,104,105], Evelin Mihailov[106], Yuri Milaneschi[49], Lili Milani[106], Francis M. Mondimore[78], Grant W. Montgomery[33], Sara Mostafavi[107,108], Niamh Mullins[56], Matthias Nauck[109,110], Bernard Ng[108], Michel G. Nivard[41], Dale R. Nyholt[111], Paul F. O'Reilly[56], Hogni Oskarsson[112], Michael J. Owen[113], Jodie N. Painter[57], Carsten Bøcker Pedersen[40,43,44], Marianne Giørtz Pedersen[40,43,44], Roseann E. Peterson[47,114], Erik Pettersson[51], Wouter J. Peyrot[49], Giorgio Pistis[55], Danielle Posthuma[115,116], Jorge A. Quiroz[117], Per Qvist[38,39,40], John P. Rice[118], Brien P. Riley[47], Margarita Rivera[56,119], Saira Saeed Mirza[66], Robert Schoevers[120], Eva C. Schulte[121,122], Ling Shen[89], Jianxin Shi[123], Stanley I. Shyn[124], Engilbert Sigurdsson[125], Grant C.B. Sinnamon[126], Johannes H. Smit[49], Daniel J. Smith[127], Hreinn Stefansson[128], Stacy Steinberg[128], Fabian Streit[76], Jana Strohmaier[76], Katherine E. Tansey[129], Henning Teismann[130], Alexander Teumer[131], Wesley Thompson[40,83,132,133], Pippa A. Thomson[132], Thorgeir E. Thorgeirsson[128], Matthew Traylor[134], Jens Treutlein[76], Vassily Trubetskoy[35], André G. Uitterlinden[135], Daniel Umbricht[136], Sandra Van der Auwera[137], Albert M. van Hemert[138], Alexander Viktorin[51], Peter M. Visscher[32,33], Yunpeng Wang[40,83,133], Bradley T. Webb[139], Shantel Marie Weinsheimer[40,83], Jürgen Wellmann[130], Gonneke Willemsen[41], Stephanie H. Witt[76], Yang Wu[32], Hualin S. Xi[140], Jian Yang[33,141], Futao Zhang[32], Volker Arolt[142], Bernhard T. Baune[45], Klaus Berger[130], Dorret I. Boomsma[41], Sven Cichon[64,75,143,144], Udo Dannlowski[142], E.J.C. de Geus[10,145], J. Raymond DePaulo[78], Enrico Domenici[146], Katharina Domschke[147], Tõnu Esko[36,106], Hans J. Grabe[137], Steven P. Hamilton[148], Caroline Hayward[149], Andrew C. Heath[118], Kenneth S. Kendler[47], Stefan Kloiber[88,150,151], Glyn Lewis[152], Qingqin S. Li[153], Susanne Lucae[88], Pamela A.F. Madden[118], Patrik K. Magnusson[51], Nicholas G. Martin[79], Andrew M. McIntosh[42,63], Andres Metspalu[106,154], Ole Mors[40,155], Preben Bo Mortensen[39,40,43,44], Bertram Müller-Myhsok[1,46,156], Merete Nordentoft[40,157], Markus M. Nöthen[64,65], Michael C. O'Donovan[113], Sara A. Paciga[158], Nancy L. Pedersen[51],

Brenda W.J.H. Penninx[49], Roy H. Perlis[68,159], David J. Porteous[160], James B. Potash[161], Martin Preisig[55], Marcella Rietschel[76], Catherine Schaefer[89], Thomas G. Schulze[76,122,162,163,164], Jordan W. Smoller[68,69,70], Kari Stefansson[155,165], Henning Tiemeier[66,166,167], Rudolf Uher[168], Henry Völzke[131], Myrna M. Weissman[102,169], Thomas Werge[40,83,170], Cathryn M. Lewis[56,171], Douglas F. Levinson[172], Gerome Breen[56,173], Anders D. Børglum[38,39,40] & Patrick F. Sullivan[51,174,175]

[32]Institute for Molecular Bioscience, The University of Queensland, Brisbane 4072 QLD, Australia. [33]Queensland Brain Institute, The University of Queensland, Brisbane 4072 QLD, Australia. [34]Analytic and Translational Genetics Unit, Massachusetts General Hospital, Boston, MA 02114, USA. [35]Department of Psychiatry and Psychotherapy, Universitätsmedizin Berlin Campus Charité Mitte, Berlin 14129, Germany. [36]Medical and Population Genetics, Broad Institute, Cambridge, MA 02142, USA. [37]Centre for Psychiatry Research, Department of Clinical Neuroscience, Karolinska Institutet, Stockholm 17177 SE, Sweden. [38]Department of Biomedicine, Aarhus University, Aarhus 8000, Denmark. [39]iSEQ, Centre for Integrative Sequencing, Aarhus University, Aarhus 8000, Denmark. [40]iPSYCH, The Lundbeck Foundation Initiative for Integrative Psychiatric Research, Aarhus 8000, Denmark. [41]Department of Biological Psychology & EMGO+ Institute for Health and Care Research, Vrije Universiteit Amsterdam, Amsterdam 1081 BT, Netherlands. [42]Division of Psychiatry, University of Edinburgh, Edinburgh EH10 5HF, UK. [43]Centre for Integrated Register-based Research, Aarhus University, Aarhus 8210, Denmark. [44]National Centre for Register-Based Research, Aarhus University, Aarhus 8210, Denmark. [45]Discipline of Psychiatry, University of Adelaide, Adelaide 5000 SA, Australia. [46]Munich Cluster for Systems Neurology (SyNergy), Munich 81377, Germany. [47]Department of Psychiatry, Virginia Commonwealth University, Richmond, VA 22903, USA. [48]Center for Neonatal Screening, Department for Congenital Disorders, Statens Serum Institut, Copenhagen 2300, Denmark. [49]Department of Psychiatry, Vrije Universiteit Medical Center and GGZ inGeest, Amsterdam 1081 NL, Netherlands. [50]Virginia Institute for Psychiatric and Behavior Genetics, Richmond, VA 23298, USA. [51]Department of Medical Epidemiology and Biostatistics, Karolinska Institutet, Stockholm 17177 SE, Sweden. [52]Department of Clinical Medicine, Translational Neuropsychiatry Unit, Aarhus University, Aarhus 8240, Denmark. [53]Human Genetics, Wellcome Trust Sanger Institute, Cambridge, CB10 1SA, UK. [54]Statistical genomics and systems genetics, European Bioinformatics Institute (EMBL-EBI), Cambridge, CB10 1 SD, UK. [55]Department of Psychiatry, University Hospital of Lausanne, Prilly, Vaud 1004, Switzerland. [56]MRC Social Genetic and Developmental Psychiatry Centre, King's College London, London WC2R 2LS, UK. [57]Genetics and Computational Biology, QIMR Berghofer Medical Research Institute, Herston 4006 QLD, Australia. [58]Centre for Advanced Imaging, The University of Queensland, Saint Lucia 4072 QLD, Australia. [59]Queensland Brain Institute, The University of Queensland, Saint Lucia 4072 QLD, Australia. [60]Psychological Medicine, Cardiff University, Cardiff CF14 4XN, UK. [61]Center for Genomic and Computational Biology, Duke University, Durham, NC 27705, USA. [62]Division of Medical Genetics, Department of Pediatrics, Duke University, Durham, NC 27708, USA. [63]Centre for Cognitive Ageing and Cognitive Epidemiology, University of Edinburgh, Edinburgh EH8 9JZ, UK. [64]Institute of Human Genetics, University of Bonn, Bonn 53127 DE, Germany. [65]Life & Brain Center, Department of Genomics, University of Bonn, Bonn 53127, Germany. [66]Epidemiology, Erasmus MC, Rotterdam 3015 Zuid-Holland, Netherlands. [67]Psychiatry, Dokuz Eylul University School Of Medicine, Izmir 35220, Turkey. [68]Department of Psychiatry, Massachusetts General Hospital, Boston, MA 02114, USA. [69]Psychiatric and Neurodevelopmental Genetics Unit (PNGU), Massachusetts General Hospital, Boston, MA 02114, USA. [70]Stanley Center for Psychiatric Research, Broad Institute, Cambridge, MA 02142, USA. [71]Neuroscience and Mental Health, Cardiff University, Cardiff CF24 4HQ, UK. [72]Bioinformatics, University of British Columbia, Vancouver V5Z 4S6 BC, Canada. [73]Department of Mathematics, Massachusetts Institute of Technology, Cambridge, MA 02142, USA. [74]Department of Psychiatry (UPK), University of Basel, Basel 4002, Switzerland. [75]Human Genomics Research Group, Department of Biomedicine, University of Basel, Basel 4031, Switzerland. [76]Department of Genetic Epidemiology in Psychiatry, Central Institute of Mental Health, Medical Faculty Mannheim, Heidelberg University, Mannheim 68159 Baden-Württemberg, Germany. [77]Department of Psychiatry, Trinity College Dublin, Dublin 8, Ireland. [78]Psychiatry & Behavioral Sciences, Johns Hopkins University, Baltimore, MD 21287, USA. [79]Genetics and Computational Biology, QIMR Berghofer Medical Research Institute, Brisbane 4006 QLD, Australia. [80]Bioinformatics Research Centre, Aarhus University, Aarhus 8000, Denmark. [81]Institute of Genetic Medicine, Newcastle University, Newcastle upon Tyne NE1 3BZ, England. [82]Danish Headache Centre, Department of Neurology, Rigshospitalet, Glostrup 2600, Denmark. [83]Institute of Biological Psychiatry, Mental Health Center Sct. Hans, Mental Health Services Capital Region of Denmark, Copenhagen 4000, Denmark. [84]iPSYCH, The Lundbeck Foundation Initiative for Psychiatric Research, Copenhagen 8000, Denmark. [85]Brain and Mind Centre, University of Sydney, Sydney 2050 NSW, Australia. [86]Interfaculty Institute for Genetics and Functional Genomics, Department of Functional Genomics, University Medicine and Ernst Moritz Arndt University Greifswald, Greifswald 17489 Mecklenburg-Vorpommern, Germany. [87]Roche Pharmaceutical Research and Early Development, Pharmaceutical Sciences, Roche Innovation Center Basel, F. Hoffmann-La Roche Ltd, Basel 4070, Switzerland. [88]Max Planck Institute of Psychiatry, Munich 80804, Germany. [89]Division of Research, Kaiser Permanente Northern California, Oakland, CA 94612, USA. [90]Psychiatry & The Behavioral Sciences, University of Southern California, Los Angeles, CA 90033, USA. [91]Department of Biomedical Informatics, Harvard Medical School, Boston, MA 02115, USA. [92]Department of Medicine, Brigham and Women's Hospital, Boston, MA 02115, USA. [93]Informatics Program, Boston Children's Hospital, Boston, MA 02115, USA. [94]Wellcome Trust Centre for Human Genetics, University of Oxford, Oxford OX3 7BN, UK. [95]Department of Endocrinology at Herlev University Hospital, University of Copenhagen, Copenhagen 2730, Denmark. [96]Institute of Social and Preventive Medicine (IUMSP), University Hospital of Lausanne, Lausanne, VD 1010, Switzerland. [97]Swiss Institute of Bioinformatics, Lausanne, VD 1015, Switzerland. [98]Division of Psychiatry, Centre for Clinical Brain Sciences, University of Edinburgh, Edinburgh EH16 4SB, UK. [99]Mental Health, NHS 24, Glasgow G12 0XH, UK. [100]Department of Psychiatry and Psychotherapy, University of Bonn, Bonn 53105, Germany. [101]Statistics, University of Oxford, Oxford OX1 3LB, UK. [102]Psychiatry, Columbia University College of Physicians and Surgeons, New York, NY 10032, USA. [103]School of Psychology and Counseling, Queensland University of Technology, Brisbane, QLD 4059, Australia. [104]Child and Youth Mental Health Service, Children's Health Queensland Hospital and Health Service, South Brisbane, QLD 4000, Australia. [105]Child Health Research Centre, University of Queensland, Brisbane, QLD 4101, Australia. [106]Estonian Genome Center, University of Tartu, Tartu 51005, Estonia. [107]Medical Genetics, University of British Columbia, Vancouver, BC V6H 3N1, Canada. [108]Statistics, University of British Columbia, Vancouver, BC V6T 1Z4, Canada. [109]DZHK (German Centre for Cardiovascular Research), Partner Site Greifswald, University Medicine, University Medicine Greifswald, Greifswald, Mecklenburg-Vorpommern 17489, Germany. [110]Institute of Clinical Chemistry and Laboratory Medicine, University Medicine Greifswald, Greifswald, Mecklenburg-Vorpommern 17489, Germany. [111]Institute of Health and Biomedical Innovation, Queensland University of Technology, Brisbane, QLD 4059, Australia. [112]Humus, Reykjavik 101, Iceland. [113]MRC Centre for Neuropsychiatric Genetics and Genomics, Cardiff University, Cardiff CF24 4HQ, UK. [114]Virginia Institute for Psychiatric & Behavioral Genetics, Virginia Commonwealth University, Richmond, VA 23298, USA. [115]Clinical Genetics, Vrije Universiteit Medical Center, Amsterdam 1081HV, Netherlands. [116]Complex Trait Genetics, Vrije Universiteit Amsterdam, Amsterdam 1081 HV, Netherlands. [117]Solid Biosciences, Boston, MA 02139, USA. [118]Department of Psychiatry, Washington University in Saint Louis

School of Medicine, Saint Louis, MO 63110, USA. [119]Department of Biochemistry and Molecular Biology II, Institute of Neurosciences, Center for Biomedical Research, University of Granada, Granada CP 18100, Spain. [120]Department of Psychiatry, University of Groningen, University Medical Center Groningen, Groningen 9700 RB, Netherlands. [121]Department of Psychiatry and Psychotherapy, Medical Center of the University of Munich, Campus Innenstadt, Munich 80336, Germany. [122]Institute of Psychiatric Phenomics and Genomics (IPPG), Medical Center of the University of Munich, Campus Innenstadt, Munich 80336, Germany. [123]Division of Cancer Epidemiology and Genetics, National Cancer Institute, Bethesda, MD 20892, USA. [124]Behavioral Health Services, Kaiser Permanente Washington, Seattle, WA 98112, USA. [125]Faculty of Medicine, Department of Psychiatry, University of Iceland, Reykjavik 101, Iceland. [126]School of Medicine and Dentistry, James Cook University, Townsville, QLD 4811, Australia. [127]Institute of Health and Wellbeing, University of Glasgow, Glasgow G12 8RZ, UK. [128]deCODE Genetics/Amgen, Reykjavik 101, Iceland. [129]College of Biomedical and Life Sciences, Cardiff University, Cardiff CF14 4EP, UK. [130]Institute of Epidemiology and Social Medicine, University of Münster, Münster, Nordrhein-Westfalen 48149, Germany. [131]Institute for Community Medicine, University Medicine Greifswald, Greifswald, Mecklenburg-Vorpommern 17489, Germany. [132]Department of Psychiatry, University of California, San Diego, San Diego, CA 92093, USA. [133]KG Jebsen Centre for Psychosis Research, Norway Division of Mental Health and Addiction, Oslo University Hospital, Oslo 0407, Norway. [134]Clinical Neurosciences, University of Cambridge, Cambridge CB2 1QW, UK. [135]Internal Medicine, Erasmus MC, Rotterdam, Zuid-Holland 3015, Netherlands. [136]Roche Pharmaceutical Research and Early Development, Neuroscience, Ophthalmology and Rare Diseases Discovery & Translational Medicine Area, Roche Innovation Center Basel, F. Hoffmann-La Roche Ltd, Basel 4070, Switzerland. [137]Department of Psychiatry and Psychotherapy, University Medicine Greifswald, Greifswald, Mecklenburg-Vorpommern 17475, Germany. [138]Department of Psychiatry, Leiden University Medical Center, Leiden 2333 ZA, Netherlands. [139]Virginia Institute of Psychiatric & Behavioral Genetics, Virginia Commonwealth University, Richmond, VA 23298, USA. [140]Computational Sciences Center of Emphasis, Pfizer Global Research and Development, Cambridge, MA 02139, USA. [141]Institute for Molecular Bioscience; Queensland Brain Institute, The University of Queensland, Brisbane, QLD 4072, Australia. [142]Department of Psychiatry, University of Münster, Münster, Nordrhein-Westfalen 48149, Germany. [143]Institute of Medical Genetics and Pathology, University Hospital Basel, University of Basel, Basel 4031, Switzerland. [144]Institute of Neuroscience and Medicine (INM-1), Research Center Juelich, Juelich 52425, Germany. [145]Amsterdam Public Health Institute, Vrije Universiteit Medical Center, Amsterdam 1081 BT, Netherlands. [146]Centre for Integrative Biology, Università degli Studi di Trento, Trento, Trentino-Alto Adige 38123, Italy. [147]Department of Psychiatry and Psychotherapy, Medical Center, University of Freiburg, Faculty of Medicine, University of Freiburg, Freiburg 79104, Germany. [148]Psychiatry, Kaiser Permanente Northern California, San Francisco, CA 94115, USA. [149]Medical Research Council Human Genetics Unit, Institute of Genetics and Molecular Medicine, University of Edinburgh, Edinburgh EH4 2XU, UK. [150]Department of Psychiatry, University of Toronto, Toronto, ON M5T 1R8, Canada. [151]Centre for Addiction and Mental Health, Toronto, ON M6J 1H4, Canada. [152]Division of Psychiatry, University College London, London W1T 7NF, UK. [153]Neuroscience Therapeutic Area, Janssen Research and Development, LLC, Titusville, NJ 08560, USA. [154]Institute of Molecular and Cell Biology, University of Tartu, Tartu 51010, Estonia. [155]Psychosis Research Unit, Aarhus University Hospital, Risskov, Aarhus 8200, Denmark. [156]University of Liverpool, Liverpool L69 3BX, UK. [157]Mental Health Center Copenhagen, Copenhagen Universtity Hospital, Copenhagen 2100, Denmark. [158]Human Genetics and Computational Biomedicine, Pfizer Global Research and Development, Groton, CT 06340, USA. [159]Psychiatry, Harvard Medical School, Boston, MA 02215, USA. [160]Medical Genetics Section, CGEM, IGMM, University of Edinburgh, Edinburgh EH4 2XU, UK. [161]Psychiatry, University of Iowa, Iowa City, IA 52246, USA. [162]Department of Psychiatry and Behavioral Sciences, Johns Hopkins University, Baltimore, MD 21287, USA. [163]Department of Psychiatry and Psychotherapy, University Medical Center Göttingen, Goettingen, Niedersachsen 37075, Germany. [164]Human Genetics Branch, NIMH Division of Intramural Research Programs, Bethesda, MD 20892-9663, USA. [165]Faculty of Medicine, University of Iceland, Reykjavik 101, Iceland. [166]Child and Adolescent Psychiatry, Erasmus MC, Rotterdam, Zuid-Holland 3015, Netherlands. [167]Psychiatry, Erasmus MC, Rotterdam, Zuid-Holland 3015, Netherlands. [168]Psychiatry, Dalhousie University, Halifax, NS B3H 2E2, Canada. [169]Division of Epidemiology, New York State Psychiatric Institute, New York, NY 10032, USA. [170]Department of Clinical Medicine, University of Copenhagen, Copenhagen 2200, Denmark. [171]Department of Medical & Molecular Genetics, King's College London, London WC2R 2LS, UK. [172]Psychiatry & Behavioral Sciences, Stanford University, Stanford, CA 94305-5717, USA. [173]NIHR BRC for Mental Health, King's College London, London SE5 8AF, UK. [174]Genetics, University of North Carolina at Chapel Hill, Chapel Hill, NC 27514, USA. [175]Psychiatry, University of North Carolina at Chapel Hill, Chapel Hill, NC 27514, USA

