## [Peer Review File · Nature Communications]

Reviewers' comments:

Reviewer #1 (Remarks to the Author):

The authors aim to address a central question about the relative contributions of genes and environment on DNA methylation variability and disease risk. The following are some specific comments about the approach and results.

- The running title does not seem appropriate
- Given that they chose tagCpGs, it would be helpful to know what the average correlations between CpGs within VMRs are to understand how valid this approach is. Why not use the average methylation value for the VMR instead? What is the advantage to using the MAD score?
- For lines 316-7, why does it matter whether models are selected with wider or narrower margins? What are the implications for this?
- The authors look at a combination of model types involving G and E to determine best models in a mixed bag of exposures and outcomes. It would be interesting to know whether the relative combination of best modeled type varies depending on the health outcome evaluated? They could easily stratify their results by disease outcome to see if GxE matters more in one disease than another. Otherwise they are implicitly assuming that the models underlying all complex diseases are the same.
- Beginning with line 385, the authors address whether variability in CpG itself can affect results regarding best models. It would be a good idea to further test whether the number of CpGs contained within/contributing to a VMR also matters as there might be a difference between a VMR defined by 2 CpGs compared to a VMR defined by 10 consecutive CpGs.
- I am a little confused at the replication results presented in lines 461-471. It looks like only half of investigated VMRs even had the same direction of effect. I don't think that stating the p-value wasn't significant helps your argument much. The authors have only 9 G x E that nominally replicate out of 515 investigated which is only a 1 % success rate. I wouldn't call this replication! This does not inspire confidence that the conclusions about which types of models drive CpG variability are sound. Given this, the conclusions presented in the discussion should be much more tempered when discussing replication. In my view, there is not much evidence here for replication of the process as a whole.
- The discussion is lengthy and should be trimmed.
- Please define MAD score in the methods section.

Reviewer #2 (Remarks to the Author):

The manuscript entitled, “Variably methylated regions in the newborn epigenome: environmental, genetic and combined influences” seeks to determine what factors (genetic, environmental, or joint gene-environment) best explain inter-individual differences in cord blood DNA methylation values. The authors identified 3,982 VMRs in their initial discovery sample: PREDO I and using AIC determined the methylation variance in 44% of the VMRs was best explained by genetic variation alone and the remaining 56% by joint gene and environmental effects. Additionally, they show VMRs differ in transcription factor, enhancer, and GWAS SNP enrichment by what type of factor(s) best explain methylation variation. The authors then look in 3 additional samples to see if the methylation variance at VMRs was explained by the G, E, G+E, or GxE models at the same proportions, i.e. is consistent with the initial PREDO I findings. They conclude that most of the variation in methylation can be explained by joint G and E effects, including additive and multiplicative across the studies.

1. While there is clearly extension of what has been done previously in terms of sample numbers and additional functional annotation/enrichment analyses, the findings are mainly consistent with the Teh et al. findings. Perhaps some clarifying or additional language about the impact of these findings on research in this field, in light of previous work, is warranted, i.e. what is new and critical to know.

2. On line 797. The authors describe taking all VMRs about the 90th percentile, as described by Ong and Holbrook but it does not appear that they assessed statistical significance of these candidate VMRs. How many of the candidate VMRs were statistically significant? Those should be the set of CpG VMRS brought forward for AIC testing.

3. I have concerns that some of the G's and E's are correlated and this could cause spurious results and make it difficult to interpret findings. It may bias towards finding more joint effects. For example, if the best SNP identified in the G model is associated with preeclampsia or preterm birth through a VMR and betamethasone is given as a treatment for high preterm birth risk (see model A below), G induces an E correlation with the VMR but E is merely correlated as a result of the G effects on the VMR as opposed to influencing the methylation levels of the VMR itself (model B below). How correlated are the G's and E's under examination?

Model A: G VMR PTB risk betamethasone (E)

Model B: G + E VMR PTB risk

4. Along the same lines as #3 above, the GxE interaction models detected could be due to statistical interaction between the G and E tested as opposed to mechanistic interaction. This is a subtle but important point given the premise the authors lead in with which is that GxE related to VMRS are mechanistic in nature. This is likely an important caveat/limitation to bring up in the discussion section.

5. It seems a bit odd to identify regions of methylation variation and then collapse the methylation variance data down to a single CpG site and assume the factors that explain methylation variation across a region is best explained by that single (greatest variance) probe. The authors assume that the CpG with the highest MAD score is representative of the entire region. I don't think it is fair to claim VMRs are best explained by the models tested via AIC when a single CpG site was tested as opposed to a value that better reflects the region. Why not identify variably methylated positions (VMPs) instead and then perform AIC for single sites? Or use a VMR area statistic that incorporates size and MAD score?

6. While the authors accurately conclude that most of the variation in methylation can be explained by joint G and E effects, including additive and multiplicative across all 4 studies, there are substantial differences with respect to the proportions explained by G alone and GxE (shown in Figure 5 and Table 1).

7. Methylation profiles are known to differ by gestational age. This should be adjusted for in the analyses. I suspect that maternal age may be related to gestational age and driving some of the observations? Perhaps gestational age may be considered an "exposure"?

8. Line 271 is confusing. The authors report 5905 PREDO I VMRs were located in genes but line 248 states that 3982 VMRs were identified in PREDO I. Did the 3982 VMRs map to 5905 genes?

9. There seem to be some inconsistencies or issues with the adjusted p-values reported in Tables S2-5. For example, Table S2 reports "p_SNP_corrected_cis_window" as the "p-value of G effect corrected for the number of tested SNPs in the specific cis window (based on Bonferroni-correction)" and "p_SNP-corrected_overall: FDR corrected p-value of G effect (corrected for the number of tested SNPs in the specific cis window and corrected for all tested tagCpGs)" which seems appropriate. However, Table S3 reports "p_E_corrected_vmrs" as "FDR corrected p-value of E effect (corrected for all tested tagCpGs)", shouldn't this be corrected for all E's tested?! The Table S3

“p_E_corrected_overall” is described as the “p_value of E effect corrected for all tested tagCpGs and all tested environments (based on Bonferroni-correction)”.

10. There are details missing in the methods section describing the AIC models that make it very difficult to tell how AIC was performed, what options were used, and how many models were run. It would be helpful to know what statistical program and which arguments were used, specific regression model details, how many models were run for each of the 4 AIC parts, and how significance thresholds were determined and applied.

Reviewer #3 (Remarks to the Author):

In this manuscript the authors perform an extensive analysis to assess the relative contributions of prenatal environmental factors and genotype on DNA methylation in offspring at birth. Even though the study had an advantage of interrogating multiple cohorts with overlapping environments, and available SNP and CpG data, it did not develop deeper insights or substantially advance the field. Also the data presentation is highly complicated, and there are concerns related to data analysis that makes it hard to ascertain replication and impact.

1. There are four cohorts with different sample sizes, and they use different DNA methylation (450k and EPIC) and SNP platforms, as well as analysis methods. The extent of heterogeneity across the cohorts warrants a careful consideration of methodologies and data reduction for cross-comparison and a reliable output. The methods section details some information but it appears that data sets from same platforms were not processed the same way for all cohorts. A table providing comparisons of data processing for DNA methylation and SNPs across the four cohorts would be helpful.
2. MAF cut-off for SNPs are either not matched or missing across different cohorts. Some use <1% while others use <5%, allowing an imbalance in SNP selection across the cohorts and potentially affecting the G, G+E and GXE analysis.
3. How were the EPIC and 450K arrays data merged? Epic arrays have a higher representation of enhancers, did that anyway influence the genomic annotation analysis.
4. All the models were adjusted only for ethnicity and gender. Why was gestational age not accounted for in the models. In cohorts like UCI the preterm criterion used was <34 wks even though the clinical criterion is <37 weeks. Differences in gestational age are known to have strong association with neonatal epigenome.

5. There is identification of VMRs in each cohort, but not a clear comparison of how many were common across the (i) cohorts and (ii) environment/disease of interest. A venn diagram of the overlaps of VMRs across the cohorts, different models and environments will provide a better idea of replication of the findings.
6. The distribution of methylation at CpGs in VMRs was unimodal for PREDO I. Was this true for rest of the cohorts? Also, how similar/different were the lengths of overlapping VMRs across the cohorts.
7. It seems only univariate analysis was tested for each environment. A multivariate analysis would be helpful to ascertain environment specific affects. For example, adjusting for smoking and maternal age would be important to ascertain maternal mood specific effects.
8. Bigger sample size usually tends to capture higher number of variable CpGs. Its strange that by using 450K platform for PREDO 1, comprising of 817 samples yielded 3,982 VMRs, while in DCHS comprising of 107 subjects yielded 6,072 VMRs. Is it due to the differences in the processing of the methylation data?
9. Its not clear if the methylation data in Drakenstein cohort was corrected for cell type?
10. Authors mention 'In PREDO and UCI, we restricted the analysis to DeepSEA variants while we used the pruned SNP-set in DCHS.' Why was DeepSEA variant analysis restricted to only two cohorts?
11. This study is a discovery analysis and the authors could have taken an advantage of the data to query both cis and trans SNP interactions and the varying cis- distances. By pre-selecting a physical distance of 1Mb limited the scope of their findings.
12. Table 1 specific queries
 - a. Why are MoBa details missing from Table 1.
 - b. Its not clear why G model was the winning model in PREDO I, while it was G X E in rest of the cohorts including PREDO II (with many overlapping environments). Did the authors try randomly selecting a subset of PREDO 1 to match the smaller sample size of other cohorts and compare the findings?
 - c. In results it is mentioned 'In total, 44 % of tagCpGs were best explained by G (n=1,759), followed294 by GxE (32%, n=1, 284) and G+E (24%, n=938) (Figure 2B).' But in table the numbers are different E=2%, G=45.1%, G+E=23.9%. GxE=29%. Which one is correct?
13. The relevance of G + E analysis is still questionable, especially in context of the redundancy in the G+E (Model 3: Methylation ~ covariates + SNP + environment) and GxE (Model 4: Methylation ~ covariates + SNP + environment + SNP x environment) models. Results from model 3 could potentially be a subset of results from model 4.
14. Data access –

a. In the absence of availability of raw and processed methylation data it is hard to assess the data qualities.

b. The open access constraints for methylation data used in this study are only mentioned for PREDO. There are no data access statements for the other cohorts.

15. There is replication of GxE and G+E combinations across MoBa and PREDO1, but their relevance in context of genes involved are not discussed.

16. The manuscript seems to be an amalgamation of multiple disconnected findings. Perhaps looking at a smaller number of overlapping environments and VMRs across cohorts with balanced sample size and SNPs would help derive more impactful and cohesive conclusions.

We thank the Editor for allowing us to submit a revised version of the manuscript and appreciate the constructive comments of the reviewers. We have addressed all methodological concerns, clarified the details of our approach and streamlined the results section and the discussion, as suggested by the reviewers. Importantly, we have rerun our analysis now covarying for gestational age, this did not substantially alter our findings. We now also present our analyses stratified by specific prenatal phenotypes. Focusing on VMRs overlapping across cohorts, we can show that the best model is highly consistent for specific VMRs. We have detailed all new analyses and findings in the manuscript and in the point-by-point responses to the reviewers below.

Highlighting the novelty as compared to a previous publication by Teh et al., was a recurrent comment of the reviewers. While building on findings presented initially by Teh et al., the access to multiple cohorts allows us to now report consistent patterns of best models across different cohorts and for specific VMRs. In our view, this significantly adds to the robustness of the initial findings. In addition, we have added an explicit distinction between GxE and G+E models (not investigated in Teh et al.). Since these two models appear to be functionally different and enriched for different disease SNPs, we feel that this is an important contribution. In addition, mapping of specific putative functional SNP variants to each VMR model now allows SNP-based functional and disease relevant analyses that were not presented in Teh et al.. In particular, these latter analyses revealed that functional GWAS annotation will benefit from including not only main effect meQTLs but also G+E and GxE models. This additional layer of annotation could lead to important novel biological insights into the pathomechanisms of common diseases in which both genetic and environmental factors are important contributors to risk.

All changes to the previous version of the manuscript are depicted in blue in the revised manuscript. All changes which we refer to in the replies to the reviewers are additionally highlighted in yellow.

Reviewers' comments:

Reviewer #1 (Remarks to the Author):

The authors aim to address a central question about the relative contributions of genes and environment on DNA methylation variability and disease risk. The following are

some specific comments about the approach and results.

- The running title does not seem appropriate

We have adapted the running title accordingly: “Combination of genetic and environmental effects best explain DNA cordblood methylation in variably methylated regions”.

- Given that they chose tagCpGs, it would be helpful to know what the average correlations between CpGs within VMRs are to understand how valid this approach is. why not use the average methylation value for the VMR instead? What is the advantage to using the MAD score?

We followed the approach by Teh et al. where each candidate VMR was represented by a tagCpG, namely the CpG with the highest MAD-score in the respective VMR. The methylation level of the tagCpG was highly correlated with the average methylation of the respective VMR (mean $r=0.85$, $sd=0.09$). We have added the following sentence to the manuscript on page 9, line 248:

“The correlation between methylation levels of tagCpG and average methylation of the respective VMR was high (mean $r=0.85$, $sd r=0.08$), suggesting that the tag CpGs are valid representatives of their VMRs.”

- For lines 316-7, why does it matter whether models are selected with wider or narrower margins? What are the implications for this?

As we chose the model with the lowest AIC-value as best model but had no further requirements on the difference of the AICs between the best model and the next best model, we wanted to assess by how much, the best model was actually the “winning” model. To make this point clearer, we have added the following to page 11, line 279:

“The delta AIC for best model G+E (mean=0.98) was also significantly higher as compared to best model G (mean=0.89, $p=2.58 \times 10^{-03}$).

GxE models thus appear to be winning by a significantly larger AIC margin over the next best model, when compared to the other types of winning models (see Figure 2C).”

- The authors look at a combination of model types involving G and E to determine best models in a mixed bag of exposures and outcomes. It would be interesting to know whether the relative combination of best modeled type varies depending on the health outcome evaluated? They could easily stratify their results by disease outcome to see if GxE matters more in one disease than another. Otherwise they are implicitly assuming that the models underlying all complex diseases are the same.

We thank the reviewer for this suggestion. Indeed, we see quite substantial differences of the relative impact of G and E on DNA methylation when stratifying by different types of E/maternal disease. In fact, maternal age and betamethasone treatment show the highest proportion of VMRs with best models G+E and GxE (about 25%), while other factors had significantly less of the best G+E or GxE models (see Figure S5B). This analysis suggests that as expected, different types of exposures or maternal factors have different relative impact on DNA methylation. However, it also supports that even for those exposures with the highest fraction of VMRs best explained by E alone, the combined models of G+E and GxE still remained the best models in an even higher fractions of VMRs. We have added a new paragraph about this analysis starting on page 14, line 353:

“Up to now, we chose the best model with regard to a multitude of different prenatal phenotypes. We next investigated if the same pattern of best models is observed across the different environmental phenotype by determining the best model (E , G, G+E and GxE) for each phenotype independently. We did not observe high correlations between the different investigated prenatal phenotypes (see Figure S5A). The strongest correlation was present between anxiety and depression scores (Pearson’s correlation coefficient $r=0.86$) followed by the area under the curve (AUC) of the oral glucose tolerance test (ogtt) and gestational diabetes ($r=0.69$). All other correlations were below

0.40. In this analysis, we observed substantial differences of the relative impact of G and E on DNA methylation when stratifying by different types of prenatal phenotypes. In fact, maternal age and betamethasone treatment show the highest proportion of VMRs with the best models G+E and GxE (about 25%), while other prenatal factors had significantly less of the best G+E or GxE models (see Figure S5B). This analysis suggests that as expected, different types of exposures or maternal factors have different relative impact on DNA methylation. However, it also supports that even for those exposures with the highest fraction of VMRs best explained by E alone, combined models of G+E and GxE remain the best models in even higher fractions of VMRs (see Figure S5B)."

- Beginning with line. 385, the authors address whether variability in CpG itself can affect results regarding best models. It would be a good idea to further test whether the number of CpGs contained within/contributing to a VMR also matters as there might be a difference between a VMR defined by 2 CpGs compared to a VMR defined by 10 consecutive CpGs.

In PREDO I, most VMRs (n=2,683 of 4,031) included 2 CpGs. We checked if VMRs with 2 CpGs differed from VMRs with more than 2 CpGs in their pattern and found that VMRs including 3 or more CpGs show significantly more best G+E models. We have added this to page 12, line 307:

"A slight enrichment for G+E models was observed in longer VMRs with at least 3 CpGs ($p=9.00 \times 10^{-06}$, Fisher-test, OR=1.39, see Figure S3)."

- I am a little confused at the replication results presented in lines 461-471. It looks like only half of investigated VMRs even had the same direction of effect. I don't think that stating the p-value wasn't significant helps your argument much. The authors have only 9 G x E that nominally replicate out of 515 investigated which is only a 1 % success rate. I wouldn't call this replication! This does not inspire confidence that the conclusions

about which types of models drive CpG variability are sound. Given this, the conclusions presented in the discussion should be much more tempered when discussing replication. In my view, there is not much evidence here for replication of the process as a whole.

We agree with the reviewer that replication of specific effects is weak and discuss that differences in exposures and specific variably methylated sites may contribute to this finding. However, when assessing those VMRs defined as VMRs with a MAD at the 90th percentile across all samples and array types – there is strong consistency of the best model for each of these VMRs across studies (see results page 15). We have now included this in the discussion on page 18, line 477:

“In addition to consistent findings using AIC-based approaches, we also observed some indication for validation of individual GxE and G+E combinations on selected VMRs using p-value based criteria, with a small number of specific G+E and GxE effects on VMRs replicating between the PREDO I and the MoBa cohort. The low number of specific replications could be due to lack of overall power as well as larger differences in prenatal factors between these two cohorts (see Table 1). As shown in Figure S5B, which specific G and E combinations best explain VMRs is also dependent on the specific prenatal factors. Larger and more homogenous cohorts regarding exposures will be needed for such analyses to be more conclusive.”

- The discussion is lengthy and should be trimmed.

We have shortened the discussion.

- Please define MAD score in the methods section.

We apologize that we have missed this and have now added to the Methods on page 28, line 730:

“For each CpG-site, the MAD (median absolute deviation) is defined as the median of the absolute deviations from each individual’s methylation beta-value at this CpG-site to the CpG’s median.”

Reviewer #2 (Remarks to the Author):

The manuscript entitled, “Variably methylated regions in the newborn epigenome: environmental, genetic and combined influences” seeks to determine what factors (genetic, environmental, or joint gene-environment) best explain inter-individual differences in cord blood DNA methylation values. The authors identified 3,982 VMRs in their initial discovery sample: PREDO I and using AIC determined the methylation variance in 44% of the VMRs was best explained by genetic variation alone and the remaining 56% by joint gene and environmental effects. Additionally, they show VMRs differ in transcription factor, enhancer, and GWAS SNP enrichment by what type of factor(s) best explain methylation variation. The authors then look in 3 additional samples to see if the methylation variance at VMRs was explained by the G, E, G+E, or GxE models at the same proportions, i.e is consistent with the initial PREDO I findings. They conclude that most of the variation in methylation can be explained by joint G and E effects, including additive and multiplicative across the studies.

1. While there is clearly extension of what has been done previously in terms of sample numbers and additional functional annotation/enrichment analyses, the findings are mainly consistent with the Teh et al. findings. Perhaps some clarifying or additional language about the impact of these findings on research in this field, in light of previous work, is warranted, i.e. what is new and critical to know.

We have now added a paragraph on the novelty of the study to the introduction on page 8, line 203:

“The main objective of the present study was to extend our knowledge of combined effects of prenatal environment and genetic factors on DNA methylation at VMRs. Specifically, this was addressed by: 1) assessing the stability of the best explanatory factors across different cohorts and whether this extends to all environmental factors, 2) dissecting differences between additive and interactive effects of gene and environment not explored in Teh et al., 3) testing whether VMRs influenced by genetic and/or environmental factors might have a different predicted impact on gene regulation and 4) evaluating the relevance of genetic variants that interact with the environment to shape the methylome for their contribution to genetic disease risk.”

2. On line 797. The authors describe taking all VMRs about the 90th percentile, as described by Ong and Holbrook but it does not appear that they assessed statistical significance of these candidate VMRs. How many of the candidate VMRs were statistically significant? Those should be the set of CpG VMRS brought forward for AIC testing.

We apologize for the misunderstanding. We defined candidate VMRs as done in Teh et al. but did not test for significance of those VMRs with regard to null areas. We rather took all candidate VMRs into consideration to increase our search space. In their paper, Ong and Holbrooke based the bootstrapping on regions containing at least 50 probes. However, in our initial PREDO I sample, no VMR contained 50 probes or more, in fact the majority of VMRs included 2 probes. The maximum size was 24 probes. Therefore, we decided to focus only on candidate VMRs. This is described in the Methods section. To make this point clearer we have added to page 9, line 239:

“We first identified candidate VMRs, defined as regions of CpG-sites showing the highest variability across all methylation sites.”

3. I have concerns that some of the G's and E's are correlated and this could cause spurious results and make it difficult to interpret findings. It may bias towards finding more joint effects. For example, if the best SNP identified in the G model is associated

with preeclampsia or preterm birth through a VMR and betamethasone is given as a treatment for high preterm birth risk (see model A below), G induces an E correlation with the VMR but E is merely correlated as a result of the G effects on the VMR as opposed to influencing the methylation levels of the VMR itself (model B below). How correlated are the G's and E's under examination?

Model A: G VMR PTB risk betamethasone (E)

Model B: G + E VMR PTB risk

We thank the reviewer for this important remark. Actually, we have to take three different correlation issues into account here: correlations between CpG-sites, correlation between genotypes and correlation between phenotypes.

With regards to correlations between CpGs, we use the tagCpGs as representatives, the correlation between the tagCpGs themselves is small (mean $r=0.03$, $sd=0.12$) and can therefore be neglected.

We have added a sentence about this to the manuscript on page 10, line 251:

“Furthermore, tagCpGs are mainly uncorrelated with each other (mean $r=0.03$, $sd=0.12$).”

Concerning correlations between genotypes, we see comparable patterns when we use pruned genotypes (at $r^2 < 0.2$) instead of DeepSEA SNPs. Therefore, we think that correlation between genotypes is not likely to corrupt the results.

With regards to correlation of phenotypes, we now present the correlation plot of the tested phenotypes (Figure S5A). We observe high correlations between depression and anxiety ($r=0.86$) as well as between the auc OGTT and gestational diabetes ($r=0.69$). All other phenotypes showed pairwise correlations of below 0.4. We have added a new paragraph about this in the manuscript on page 14, line 356:

“We did not observe high correlations between the different investigated prenatal phenotypes (see Figure S5A). The strongest correlation was present between anxiety and depression scores (Pearson's correlation coefficient $r=0.86$) followed by the area

under the curve (AUC) of the oral glucose tolerance test (ogtt) and gestational diabetes ($r=0.69$). All other correlations were below 0.40. “

4. Along the same lines as #3 above, the GxE interaction models detected could be due to statistical interaction between the G and E tested as opposed to mechanistic interaction. This is a subtle but important point given the premise the authors lead in with which is that GxE related to VMRS are mechanistic in nature. This is likely an important caveat/limitation to bring up in the discussion section.

We agree with the reviewer and have now added the following to the discussion on page 22, line 577:

“Fourth, all reported interactions are statistical interactions and limited to a cis window around the CpG-site. Further experiments are required to assess whether these would also reflect biological/mechanistic interactions.”

5. It seems a bit odd to identify regions of methylation variation and then collapse the methylation variance data down to a single CpG site and assume the factors that explain methylation variation across a region is best explained by that single (greatest variance) probe. The authors assume that the CpG with the highest MAD score is representative of the entire region. I don't think it is fair to claim VMRs are best explained by the models tested via AIC when a single CpG site was tested as opposed to a value that better reflects the region. Why not identify variably methylated positions (VMPs) instead and then perform AIC for single sites? Or use a VMR area statistic that incorporates size and MAD score?

Indeed, focusing on VMPs is another interesting approach. However, the methylation levels of the tagCpG are highly correlated with the average methylation of the VMR (mean $r=0.85$, $sd=0.09$), suggesting that the tag CpGs are valid representatives of their VMRs. We have added on page 9, line 248:

“The correlation between methylation levels of tagCpG and average methylation of the respective VMR was high (mean $r=0.85$, sd $r=0.08$), suggesting that the tag CpGs are valid representatives of their VMRs.”

6. While the authors accurately conclude that most of the variation in methylation can be explained by joint G and E effects, including additive and multiplicative across all 4 studies, there are substantial differences with respect to the proportions explained by G alone and GxE (shown in Figure 5 and Table 1).

We agree with this comment, especially PREDO I differs in the pattern and presents with a higher proportion of best G models as compared to the other cohorts. In the updated analyses, we included gestational age as a covariate and observed that in this analysis the distribution of best models, PREDO I is more similar to the other cohorts, but still shows the highest amount of best G models.

As reviewer 3 suggested, we analyzed a subset of PREDO I of $n=150$ and reran our analysis. Indeed, in this smaller subsample, the distribution was more similar to other cohorts (<1% E, 8% G, 31% G+E, 60% GxE) suggesting that the higher fraction of best G models may come from the substantially larger sample size in PREDO I.

We have added this to the discussion on page 22, line 582:

“Fifth, as summarized in Table 1, results presented are based on cohorts which differ in ethnicity, assessed phenotypes, methylation and SNP arrays, processing pipelines and sample sizes. While all these factors may contribute to differences in the proportions of models across the cohorts (as shown for sample size by subsampling with PREDO I, data not shown), it also suggests that our findings are quite robust to these methodological issues.”

7. Methylation profiles are known to differ by gestational age. This should be adjusted for in the analyses. I suspect that maternal age may be related to gestational age and

driving some of the observations? Perhaps gestational age may be considered an “exposure”?

This is an important point. We now include gestational age as a covariate in all analyses and have adapted all results and the enrichment analysis. This did not substantially change the results. We observed only a low correlation between maternal age and gestational age in the initial PREDO I sample ($r=0.02$).

8. Line 271 is confusing. The authors report 5905 PREDO I VMRs were located in genes but line 248 states that 3982 VMRs were identified in PREDO I. Did the 3982 VMRs map to 5905 genes?

We apologize for the confusion. To streamline the manuscript, we present this analysis now in the Supplemental Results. We identified 3,982 VMRs overall, each of these VMRs includes several CpG-sites. 5,095 is the number of CpGs which are located in VMRs and which are genic. We now clarify this on page 1, line 24 of the Supplemental Results:

“As the analysis presented by Bonder et al. was based on CpGs located in proximity to the TSS of the specific transcript, we used only PREDO I CpGs located in VMRs and also located within genes ($n=5,905$).”

9. There seem to be some inconsistencies or issues with the adjusted p-values reported in Tables S2-5. For example, Table S2 reports “p_SNP_corrected_cis_window” as the “p-value of G effect corrected for the number of tested SNPs in the specific cis window (based on Bonferroni-correction)” and “p_SNP-corrected_overall: FDR corrected p-value of G effect (corrected for the number of tested SNPs in the specific cis window and corrected for all tested tagCpGs)” which seems appropriate. However, Table S3 reports “p_E_corrected_vmrs” as “FDR corrected p-value of E effect (corrected for all tested tagCpGs)”, shouldn’t this be corrected for all E’s tested?! The Table S3 “p_E_corrected_overall” is described as the “p_value of E effect corrected for all tested tagCpGs and all tested environments (based on Bonferroni-correction)”.

We apologize for the inconsistencies and have removed them. We have rerun all analyses using gestational age as an additional covariate and present the updated results. We consistently used Bonferroni correction to correct for the number of tested Es (E model), of tested cis SNPs (G model) or of tested cis SNP-environment combinations (G+E/GxE model). Afterwards we used FDR to correct for all tested tagCpGs (all models).

10. There are details missing in the methods section describing the AIC models that make it very difficult to tell how AIC was performed, what options were used, and how many models were run. It would be helpful to know what statistical program and which arguments were used, specific regression model details, how many models were run for each of the 4 AIC parts, and how significance thresholds were determined and applied.

We apologize if this was unclear. Within the Methods section, we describe in the paragraph *Regression analysis*, the programme we used, which covariates were included and how the AIC was calculated - see page 36, line 949 (please also see Figure 1):

“Regression analysis

Linear regression analysis was conducted using the *lm* function in R 3.3.1

(<https://www.r-project.org>). We included the child’s sex, gestational age, seven estimated cell counts as well as the first two (PREDO I and PREDO II), first three (UCI) and first five (DCHS I and II) principal components of the MDS analysis on the genotypes in the model. The corresponding plot of the first ten MDS-components in PREDO is depicted in Figure S4. SNP genotypes were recoded into a count of 0, 1 or 2 representing the number of minor allele copies. For each VMR site, we tested SNPs located in a 1MB window up- and downstream of the specific site. In PREDO and UCI, we restricted the analysis to DeepSEA variants while we used the pruned SNP-set in DCHS.

For each VMR, we tested four models:

- (1) Methylation at tagCpG ~ covariates + environment
- (2) Methylation at tagCpG ~ covariates + SNP

(3) Methylation at tagCpG \sim covariates + SNP + environment

(4) Methylation at tagCpG \sim covariates + SNP + environment + SNP x environment

In model (1) we included all ten different environments, in model (2) all DeepSEA cis SNPs and in models (3) and (4) all possible environment-cis-SNP combinations. Please see Figure 1.

For each model, the AIC, Akaike's information criterion ⁴³ was calculated and the model with the lowest AIC was chosen as the best model. The AIC was obtained using the *AIC* function in R 3.3.1 (<https://www.r-project.org>).

P-values were obtained from the summary function and adjusted for the number of tested Es (E model), of tested cis SNPs (G model) or of tested cis SNP-environment combinations (G+E/GxE model) using Bonferroni-correction. Afterwards, we used FDR to correct for all tested tagCpGs (all models) using *p.adjust* in R. "

Reviewer #3 (Remarks to the Author):

In this manuscript the authors perform an extensive analysis to assess the relative contributions of prenatal environmental factors and genotype on DNA methylation in offspring at birth. Even though the study had an advantage of interrogating multiple cohorts with overlapping environments, and available SNP and CpG data, it did not develop deeper insights or substantially advance the field. Also the data presentation is highly complicated, and there are concerns related to data analysis that makes it hard to ascertain replication and impact.

1. There are four cohorts with different sample sizes, and they use different DNA methylation (450k and EPIC) and SNP platforms, as well as analysis methods. The extent of heterogeneity across the cohorts warrants a careful consideration of methodologies and data reduction for cross-comparison and a reliable output. The methods section details some information but it appears that data sets from same platforms were not

processed the same way for all cohorts. A table providing comparisons of data processing for DNA methylation and SNPs across the four cohorts would be helpful.

We agree that heterogeneity is an important issue. While we ran a standardized analysis on all cohorts, methylation quality control and imputation was carried out center-specifically. We have added cohort-specific information to Table 1 and also added a paragraph on the heterogeneity to the discussion on page 22 line 582:

“Fifth, as summarized in Table 1, results presented are based on cohorts which differ in ethnicity, assessed phenotypes, methylation and SNP arrays, processing pipelines and sample sizes. While all these factors may contribute to differences in the proportions of models across the cohorts (as shown for sample size by subsampling with PREDO I, data not shown), it also suggests that our findings are quite robust to these methodological issues.”

2. MAF cut-off for SNPs are either not matched or missing across different cohorts. Some use <1% while others use <5%, allowing an imbalance in SNP selection across the cohorts and potentially affecting the G, G+E and GXE analysis.

We apologize for the confusion. While MAF-inclusion differed for quality control before imputation, for all cohorts, only SNPs with MAF >1% were used in the analysis based on imputed genotypes. We have added this information to the Methods section for each cohort.

3. How were the EPIC and 450K arrays data merged? Epic arrays have a higher representation of enhancers, did that anyway influence the genomic annotation analysis.

We did not merge EPIC and 450K data but rather analyzed each cohort and chip type on its own. The annotation of the analysis is always with regards to the specific methylation array as background. For joint analyses, we only used VMRs present on both arrays. For the functional annotations, we only used PREDO I data based on the 450K array as this was the largest, most homogenous cohort.

4. All the models were adjusted only for ethnicity and gender. Why was gestational age not accounted for in the models. In cohorts like UCI the preterm criterion used was <34 wks even though the clinical criterion is <37 weeks. Differences in gestational age are known to have strong association with neonatal epigenome.

We thank the reviewer for bringing up this important point. We have reconsidered our models and included gestational age as a covariate in all analyses. We used gestational age as quantitative covariate and did not use preterm birth as categorical phenotype. This new analysis did not substantially alter our results.

5. There is identification of VMRs in each cohort, but not a clear comparison of how many were common across the (i) cohorts and (ii) environment/disease of interest. A venn diagram of the overlaps of VMRs across the cohorts, different models and environments will provide a better idea of replication of the findings.

We thank the Reviewer for this comment. We have included a paragraph on the overlap of tagCpGs and consistency of best models across the cohorts on page 15, line 397:

“Overall, 387 tag CpGs overlapped between PREDO I, PREDO II, DCHS I and DCHS II (see Figure S7), which allowed us to test the consistency of the best models for specific VMRs across the different cohorts. Over 70% of the overlapping tagCPGs showed consistent best models in at least 3 cohorts (see Figure 6) with GxE being the most consistent model (for over 60% of consistent models). Focusing only on EPIC data (PREDO II, DCHSII and UCI), we identified more, namely 2,091, tag CpGs that overlap across the three cohorts and here 86% show a consistent best model in at least two of the three cohorts, despite differences in study design, prenatal phenotypes and ethnicity. Thus, the additional cohorts not only showed a consistent replication of the proportion of the models best explaining variance of VMRs but also consistency of the best model for specific VMRs. “

6. The distribution of methylation at CpGs in VMRs was unimodal for PREDO I. Was this true for rest of the cohorts? Also, how similar/different were the lengths of overlapping VMRs across the cohorts.

We saw the same pattern across the cohorts. We now show the respective plots for all cohorts (see Figure S6) on page 3, line 62 of the Supplemental Results:

“In all these cohorts, we observed the same distribution of median methylation levels at VMR CpG-sites as in PREDO I: while overall methylation levels at CpGs were bimodally distributed as expected, the distribution of methylation levels at CpGs within VMRs was unimodal and VMRs presented with intermediate methylation levels (see Figure S6).

The length of VMRs was similar across cohorts with an overall mean of 3.8 CpGs and individual means: PREDO I=3.26, PREDO II=3.36, DCHS I=3.93, DCHS II=3.66, UCI=3.57.”

7. It seems only univariate analysis was tested for each environment. A multivariate analysis would be helpful to ascertain environment specific effects. For example, adjusting for smoking and maternal age would be important to ascertain maternal mood specific effects.

We ran univariate analysis as we were interested in each E separately. We agree that for specific environmental effects, adjusting for additional phenotypes such as maternal smoking or maternal age is important. We reran models for depression and anxiety in PREDO I and used maternal age as covariate. In fact, we observed a high correlation between p-values of the unadjusted and the adjusted model ($r=0.99$). However, for specific environmental effects, more detailed analyses would be important, as now stated in the Discussion on page 22, line 582, but is beyond the scope of this manuscript:

“Additional inclusion of further covariates such as maternal smoking or maternal age may further modify the effects of specific Es but is beyond the scope of this manuscript.”

8. Bigger sample size usually tends to capture higher number of variable CpGs. Its strange that by using 450K platform for PREDO 1, comprising of 817 samples yielded 3,982 VMRs, while in DCHS comprising of 107 subjects yielded 6,072 VMRs. Is it due to the differences in the processing of the methylation data?

We agree that this finding is somewhat counterintuitive. It maybe related to differences in data processing, but more likely, the ethnically more diverse background in DCHS, as VMRs also reflect genetic heterogeneity.

We have added this to the Discussion on page 22, line 582:

“Fifth, as summarized in Table 1, results presented are based on cohorts which differ in ethnicity, assessed phenotypes, methylation and SNP arrays, processing pipelines and sample sizes. While all these factors may contribute to differences in the proportions of models across the cohorts (as shown for sample size by subsampling with PREDO I, data not shown), it also suggests that our findings are quite robust to these methodological issues.”

9. Its not clear if the methylation data in Drakenstein cohort was corrected for cell type?

We regret that we omitted this information. The Drakenstein methylation data were also corrected for cell types. We have now clarified this in the Methods on page 31, line 811:

“Neonatal blood cell counts were estimated for seven cell types: nucleated red blood cells, granulocytes, monocytes, natural killer cells, B cells, CD4(+)T cells, and CD8(+)T cells.”

10. Authors mention ‘In PREDO and UCI, we restricted the analysis to DeepSEA variants while we used the pruned SNP-set in DCHS.’ Why was DeepSEA variant analysis restricted to only two cohorts?

As too few DeepSEA variants were available per VMR in DCHS, we used pruned SNPs here. While we stated this in the Methods section, we have now explained this again in the Supplemental Results on page 3, line 62:

“As after imputation only few DeepSEA variants were available for the DCHS cohort, we performed LD-pruning in this cohort and ran the analysis on the pruned SNP set (see Methods). “

11. This study is a discovery analysis and the authors could have taken an advantage of the data to query both cis and trans SNP interactions and the varying cis- distances. By pre-selecting a physical distance of 1Mb limited the scope of their findings.

We agree that focusing on 1 MB limits the scope of the findings, however, we feel that we do not have sufficient power in our samples to increase the Cis window size or analyze trans effects. For this, larger consortia are necessary. The PREDO cohort is part of the GoDMC consortium with nearly 30,000 samples in which trans meQTL effects are studied. We have added on page 22, line 577:

“Fourth, all reported interactions are statistical interactions and limited to a *cis* window around the CpG-site. Further experiments are required to assess whether these would also reflect biological/mechanistic interactions. Much larger cohorts will be needed to assess potential *trans* effects. “

12. Table 1 specific queries

a. Why are MoBa details missing from Table 1.

We apologize that we missed this and have now added MoBa details as well as details on genotyping, methylation arrays and preprocessing to Table 1.

b. Its not clear why G model was the winning model in PREDO I, while it was G X E in rest of the cohorts including PREDO II (with many overlapping environments). Did the authors try randomly selecting a subset of PREDO 1 to match the smaller sample size of other cohorts and compare the findings?

As suggested by the Reviewer, we took a subset of PREDO I of n=150 and reran the pipeline. Indeed, in this smaller sample, PREDO I was more similar to other cohorts (<1% E, 8% G, 31% G+E, 60% GxE) suggesting that we have more power to detect G only effects in larger samples. In addition, controlling for gestational age also reduced the fraction of best G models from 46% to 30%. We have added this to the Discussion on page 22, line 582:

“Fifth, as summarized in Table 1, results presented are based on cohorts which differ in ethnicity, assessed phenotypes, methylation and SNP arrays, processing pipelines and sample sizes. While all these factors may contribute to differences in the proportions of models across the cohorts (as shown for sample size by subsampling with PREDO I, data not shown), it also suggests that our findings are quite robust to these methodological issues.”

c. In results it is mentioned ‘In total, 44 % of tagCpGs were best explained by G (n=1,759), followed294 by GxE (32%, n=1, 284) and G+E (24%, n=938) (Figure 2B).’ But in table the numbers are different E=2%, G=45.1%, G+E=23.9%. GxE=29%. Which one is correct?

Please note that the table is based on the DeepSEA results, while at the beginning of the text on page 10, line 258 we are referring to the pruned dataset.

13. The relevance of G + E analysis is still questionable, especially in context of the redundancy in the G+E (Model 3: Methylation ~ covariates + SNP + environment) and

GxE (Model 4: Methylation ~ covariates + SNP + environment + SNP x environment) models. Results from model 3 could potentially be a subset of results from model 4.

We explicitly checked if GxE models show a smaller AIC than G+E models, hence, although the best G+E models are a subset of the GxE models, they perform better with regard to AIC than if the SNP x environment term is added.

Looking at the deltaAIC, we observe that GxE models win by larger margins as opposed to G+E models. Furthermore, GxE and G+E models appear to map to functionally different genomic sites.

14. Data access –

a. In the absence of availability of raw and processed methylation data it is hard to assess the data qualities.

We agree that availability of raw data is very important and regret that due to ethical restrictions, we cannot make them directly publicly available. However, interested researchers can apply for access to the data as stated in the data access statements.

b. The open access constraints for methylation data used in this study are only mentioned for PREDO. There are no data access statements for the other cohorts.

We thank the Reviewer for pointing this out, and have added contacts for the other cohorts.

15. There is replication of GxE and G+E combinations across MoBa and PREDO1, but their relevance in context of genes involved are not discussed.

We have now included more information on the top GxE and G+E findings in the figure legends but not in the discussion, as we tried to condense the latter part of the manuscript to accommodate Reviewer 1's comment.

16. The manuscript seems to be an amalgamation of multiple disconnected findings. Perhaps looking at a smaller number of overlapping environments and VMRs across

cohorts with balanced sample size and SNPs would help derive more impactful and cohesive conclusions.

Overall, we have tried to make the manuscript flow better and have better highlighted the main findings. We hope the Reviewer finds the manuscript to be more coherent.

REVIEWERS' COMMENTS:

Reviewer #1 (Remarks to the Author):

the authors have successfully addressed my comments.

Reviewer #2 (Remarks to the Author):

Thank you for the thoughtful revisions and responses. The manuscript is much improved.

Reviewer #3 (Remarks to the Author):

The authors have revised the manuscript and provided some of the requested details and analysis. Unfortunately the new edits still leave ambiguity in data presentation that makes it difficult to confirm the replication (the most important component of the paper) of the findings and advancement in the field.

Figure 6 lumps up all the models to indicate replication of tag CpGs across the cohorts. A more useful representation (as requested earlier) would have been to plot or provide a table indicating the percentage/number of tagCpGs replicated within each model across the cohorts. The authors do provide some details in the text 'Over 70% of the overlapping tagCPGs showed consistent best models in at least 3 cohorts (see Figure 6) with GxE being the most consistent model (for over 60% of consistent models).' There seems to be an overrepresentation of GxE model in replication, does this mean that the GxE findings are more reliable than other models. Providing % replication for VMRs within each model would have brought in more clarity to the results and the conclusions presented.

Minor points

1. Table 2 – sum of the proportions explained by each model adds up to 110% for PREDO1. Suggest cross-checking the calculations. Should these proportions match those presented in Figure 2A?

2. Figure 5 legend for G+E and GxE is truncated in the submitted pdf.

We thank the Editor for allowing us to submit a final revision of the manuscript and appreciate the constructive comments of the reviewers.

Reviewers' comments:

Reviewer #1 (Remarks to the Author):

the authors have successfully addressed my comments.

We thank the reviewer for the comments which helped to improve the manuscript.

Reviewer #2 (Remarks to the Author):

Thank you for the thoughtful revisions and responses. The manuscript is much improved.

We appreciate the reviewer's comments which supported us in improving the manuscript.

Reviewer #3 (Remarks to the Author):

The authors have revised the manuscript and provided some of the requested details and analysis. Unfortunately the new edits still leave ambiguity in data presentation that makes it difficult to confirm the replication (the most important component of the paper) of the findings and advancement in the field.

Figure 6 lumps up all the models to indicate replication of tag CpGs across the cohorts. A more useful representation (as requested earlier) would have been to plot or provide a table indicating the percentage/number of tagCpGs replicated within each model across the cohorts. The authors do provide some details in the text 'Over 70% of the overlapping tagCPGs showed consistent best models in at

least 3 cohorts (see Figure 6) with GxE being the most consistent model (for over 60% of consistent models).’ There seems to be an overrepresentation of GxE model in replication, does this mean that the GxE findings are more reliable than other models. Providing % replication for VMRs within each model would have brought in more clarity to the results and the conclusions presented.

We agree with the reviewer that a more detailed description of the consistency of models is helpful. We now present the results stratified by models in

Supplementary Figure 8 (see below). Further we added at p.15, l.475:

“Within this context, we observed the GxE models were the most consistent models across the cohorts (see supp Fig 8), with 85% of the CpGs with consistent models across 5 cohorts having GxE as the best model. “

Minor points

1. Table 2 – sum of the proportions explained by each model adds up to 110% for PREDO1. Suggest cross-checking the calculations. Should these proportions match those presented in Figure 2A?

We apologize for this typo and now corrected the specific numbers. The proportions match those presented in Figure 5.

2. Figure 5 legend for G+E and GxE is truncated in the submitted pdf.

We apologize for any inconvenience and corrected this.